# *Candida albicans* pathways that protect against organic peroxides and lipid peroxidation

**Kara A. Swenson**[1], **Kyunghun Min**[2], **James B. Konopka**[1]*

**1** Department of Microbiology and Immunology, Stony Brook University, Stony Brook, New York, United States of America, **2** Department of Plant Science, Gangneung-Wonju National University, Gangneung, Republic of Korea

* james.konopka@stonybrook.edu

**Data Availability Statement:** The RNAseq data are available as S5 Table. The RNAseq data has been uploaded to the Sequence Read Archive of the National Library of Medicine of the National Institutes of Health under BioProject accession

## Abstract

Human fungal pathogens must survive diverse reactive oxygen species (ROS) produced by host immune cells that can oxidize a range of cellular molecules including proteins, lipids, and DNA. Formation of lipid radicals can be especially damaging, as it leads to a chain reaction of lipid peroxidation that causes widespread damage to the plasma membrane. Most previous studies on antioxidant pathways in fungal pathogens have been conducted with hydrogen peroxide, so the pathways used to combat organic peroxides and lipid peroxidation are not well understood. The most well-known peroxidase in *Candida albicans*, catalase, can only act on hydrogen peroxide. We therefore characterized a family of four glutathione peroxidases (GPxs) that were predicted to play an important role in reducing organic peroxides. One of the GPxs, Gpx3 is also known to activate the Cap1 transcription factor that plays the major role in inducing antioxidant genes in response to ROS. Surprisingly, we found that the only measurable role of the GPxs is activation of Cap1 and did not find a significant role for GPxs in the direct detoxification of peroxides. Furthermore, a *CAP1* deletion mutant strain was highly sensitive to organic peroxides and oxidized lipids, indicating an important role for antioxidant genes upregulated by Cap1 in protecting cells from organic peroxides. We identified *GLR1* (Glutathione reductase), a gene upregulated by Cap1, as important for protecting cells from oxidized lipids, implicating glutathione utilizing enzymes in the protection against lipid peroxidation. Furthermore, an RNA-sequencing study in *C. albicans* showed upregulation of a diverse set of antioxidant genes and protein damage pathways in response to organic peroxides. Overall, our results identify novel mechanisms by which *C. albicans* responds to oxidative stress resistance which open new avenues for understanding how fungal pathogens resist ROS in the host.

## Author summary

Human fungal pathogens must survive attack by diverse reactive oxygen species (ROS) produced by host immune cells in order to cause a lethal disseminated infection. This oxidative stress damages many different cellular molecules including proteins, lipids, and

number GSE275502. S8 Table contains all of the values from the quantitative experiments reported in this manuscript.

**Funding:** This research was supported by a Public Health Service grants awarded to J.B.K. from the National Institutes of Health (R01AI047837 and R01AI177553). K.A.S. was supported in part by a National Institutes of Health training grant (T32AI007539) from the National Institute of Allergy and Infectious Diseases. The funders had no role in the study design, data collection and analysis, decision to publish, or preparation of the manuscript.

**Competing interests:** The authors have declared that no competing interests exist.

DNA. Fungal plasma membrane lipids are one of the first targets to encounter this oxidative damage, so our studies focused on identifying the antioxidant pathways in the human fungal pathogen *Candida albicans* that protect this essential barrier. Furthermore, we included the analysis of organic (carbon-containing) peroxides, which behave differently than the more commonly studied hydrogen peroxide. Our results indicate that resistance to organic peroxides in *C. albicans* is mediated by the transcription factor Cap1 that induces the expression of a wide array of antioxidant genes. Many genes contribute to resistance to oxidation, but our studies revealed a key role for the Cap1-regulated gene *GLR1*. Altogether, these results identify novel mechanisms by which *C. albicans* protects against oxidative stress, which open new avenues for development of novel therapeutic strategies to combat fungal infections.

## Introduction

Successful pathogens require the ability to survive the stressful host environment. From the onset of infection, pathogens are bombarded by numerous stressors, including temperature changes, nutrient deprivation, osmotic/cationic stress, and oxidation [1–6]. These various forms of stress significantly narrow the number of microorganisms that can successfully replicate and cause disease in humans. For example, of the millions of fungal species only a few hundred have been found to infect humans [7]. Currently, the rates of fungal infections continue to rise, partly due to new medical procedures that provide sites for biofilm formation and immunosuppressive treatments which increase the number of immunocompromised individuals in the population [8]. In addition, the lack of research on fungal pathogens and the dearth of effective antifungal drugs heavily contribute to the alarmingly high mortality rates of invasive fungal infections [9,10]. A better understanding of how fungi survive the stressful human host environment is essential for developing new strategies for combating the increasing threat of fungal pathogens. Our work focuses on the stress resistance mechanisms of the common systemic human fungal pathogen *Candida albicans*, which is considered a critical health threat by the World Health Organization [11–13].

One main source of stress for invading *C. albicans* is the innate immune system. Briefly, innate immune cells respond to fungal pathogen-associated molecular patterns and attack fungal cells by phagocytosis or by the formation of Neutrophil Extracellular Traps during which fungal cells are exposed to many types of cellular stress, including oxidative stress [14–17]. The major source of oxidative stress comes from NADPH oxidase in the phagosome of macrophages and neutrophils which generates highly reactive superoxides that are then converted to hydrogen peroxide ($H_2O_2$) and other reactive oxygen species (ROS) [18–20]. $H_2O_2$ can be acted on by myeloperoxidase to form hypochlorous acid (bleach), which occurs at a high rate in neutrophils [3,21]. Furthermore, metal ions are pumped into the phagosome that are redox active and create a more diverse array of ROS [2,22]. ROS can also react with many cellular compounds including proteins, lipids, and DNA [23]. However, most research on oxidative stress in *C. albicans* has focused on the responses to $H_2O_2$ [1,24]. While this research is foundational and physiologically relevant, it does not encompass the breadth of organic and inorganic ROS that pathogens must combat during infection. Furthermore, while it is known that catalase can act on $H_2O_2$, it does not act on organic peroxides, which has left a gap in understanding how cells respond to other types of peroxides, such as lipid peroxides.

*C. albicans* has evolved defenses to counteract ROS from various sources, including their endogenous metabolic pathways [24]. The most established antioxidant pathway in *C. albicans*

is the detoxification of superoxide which is converted to $H_2O_2$ by superoxide dismutases (SODs), and then $H_2O_2$ is converted to water by catalase (Cat1) [25–27]. Two other major groups of antioxidant proteins are the glutathione and thioredoxin systems, which rely on the reducing potential of Cys residues on the small molecule glutathione and the protein thioredoxin, respectively [28]. The most important regulator of antioxidant genes is the AP-1 like transcription factor, Cap1. Understanding of Cap1 function has been aided by studies on the orthologous Yap1 transcription factor in *Saccharomyces cerevisiae* [29–31]. In both organisms, exposure to oxidative stress leads to the oxidation of a Cys residue on the glutathione peroxidase, Gpx3, which then forms an intermolecular disulfide bond with Cap1. This intermediate step is followed by formation of an intramolecular disulfide bridge between two Cap1 Cys residues and the release of Gpx3 (Fig 1A) [31–35]. This disulfide bridge in Cap1 causes a conformational change that masks a nuclear export signal, leading to Cap1 accumulation in the nucleus [36]. Cap1 then upregulates the transcription of many antioxidant genes in *C. albicans*, including genes that code for glutathione reductase, thioredoxin, thioredoxin reductase, and catalase [37–39]. The other major stress regulator in *C. albicans* is the Hog1 stress activated protein kinase (SAPK), but it appears to play a lesser role in oxidative stress resistance [40,41].

The front line of attack for oxidative stress during infection is the *C. albicans* plasma membrane [2,3,42,43]. ROS attacks on membrane lipids form lipid radicals, which can then react with molecular oxygen to form highly reactive lipid peroxyl radicals. These radical lipids then go on to react with other membrane lipids to propagate the radical reaction, leading to the formation of numerous lipid radicals and peroxides and resulting in widespread membrane damage and loss of integrity [44–46]. The resulting lipid peroxides can also degrade into toxic products, including aldehydes [47]. This radical chain reaction is known as lipid peroxidation and can occur in all membranes. Lipid peroxidation is a topic of much interest in mammalian cells where regulation of cell death from high levels of oxidized lipids (ferroptosis) is being investigated as a potential cancer therapy [48,49]. However, lipid peroxidation has been understudied in fungal pathogens. One potential reason for this lack of research is that the model yeast *S. cerevisiae* lacks polyunsaturated fatty acids (PUFAs), which are highly susceptible to lipid peroxidation, whereas about 30% of *C. albicans* fatty acids are polyunsaturated [2,50–52]. PUFAs are more prone to radicalization due to their conjugated double bonds which allow them to stabilize radical electrons through resonance [45].

A family of quinone oxidoreductases known as flavodoxin-like proteins (FLPs) were identified as the first family of proteins known to protect against lipid peroxidation in *C. albicans* [53]. The FLPs are thought to act by reducing ubiquinone to ubiquinol, allowing it to reduce lipid radicals in the plasma membrane. The FLPs were found to be essential for virulence in a murine infection model, further emphasizing the importance of protecting membrane lipids from oxidation during infection [54,55]. However, it is still unknown which other proteins are involved in detoxifying oxidized membrane lipids, as this has not been well characterized. One protein family predicted to protect against lipid peroxidation in *C. albicans* is the glutathione peroxidases (GPx), which reduce peroxides, including organic peroxides, to more stable alcohols (Fig 1B). This model for GPx function is based on studies in mammalian cells and *S. cerevisiae*, and work in *C. albicans* has also suggested a role for the GPxs in oxidative stress resistance [28,36,56–58]. Therefore, the goal of this study was to characterize each of the four *C. albicans* GPxs to establish their role in resistance to organic peroxides. Surprisingly, we found that the predominant function of the GPxs is Cap1 activation; the GPxs did not appear to play a critical role in the direct detoxification of peroxides. Instead, Cap1 was found to be highly important for resistance to a broad range of peroxides. Analysis of a selected set of Cap1 regulated antioxidant genes revealed that glutathione reductase (Glr1) is important for resistance to lipid peroxidation, implicating glutathione utilizing enzymes in reducing oxidized

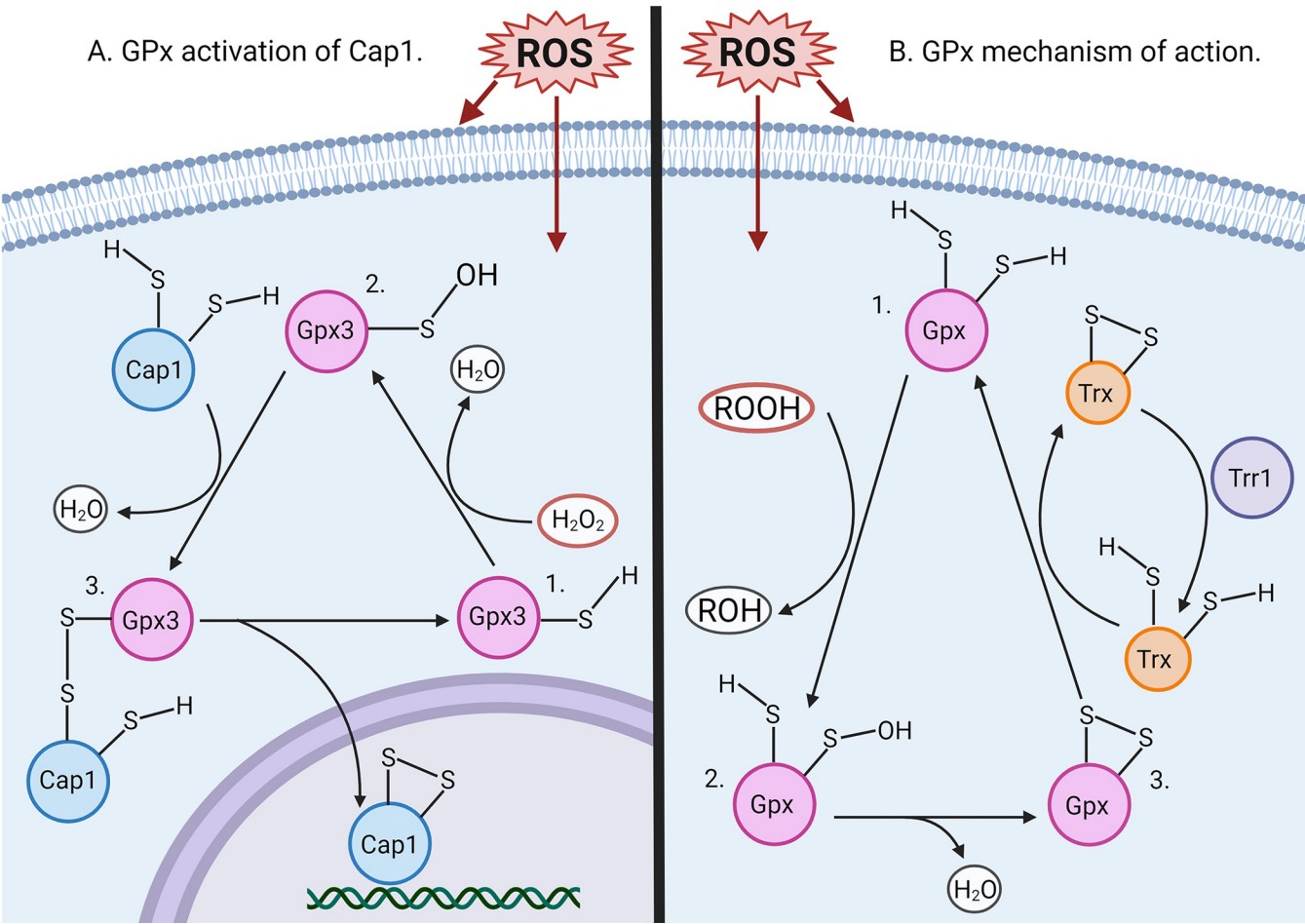

**Fig 1. Model for GPx and Cap1 mechanisms of action.** (A) ROS oxidize a Cys residue on Gpx3 which goes on to form a disulfide bridge with a Cys on Cap1. This intermediate is resolved by Cap1 forming an intracellular disulfide bridge between two of its own Cys residues leading to the release of Gpx3. The disulfide bridge in Cap1 causes a conformational change that masks a nuclear export signal on Cap1, allowing it to accumulate in the nucleus and upregulate the transcription of antioxidant genes. (B) Glutathione peroxidase (Gpx) proteins utilize a Cys residue to reduce peroxides to alcohols. The oxidized Cys forms an intramolecular disulfide bond with another Cys residue and then thioredoxin (Trx) reduces the Gpx, so it can go on to reduce another peroxide. Thioredoxin is reduced by a thioredoxin reductase (Trr), which, in turn, relies on the reducing potential of NADPH (not pictured). Figure created using BioRender.com.

lipids or their subsequent metabolites. Based on these findings, we propose a new model for protections against lipid peroxidation.

## Results

### *C. albicans* glutathione peroxidases localize to the cytoplasm and plasma membrane

Glutathione peroxidases were first identified in fungi over 30 years ago and foundational studies on their function were conducted in the model organism, *S. cerevisiae*, which has three highly conserved GPxs [56,59,60]. *C. albicans* has four GPxs, which are also highly conserved. The glutathione peroxidases in both organisms are all relatively small proteins and have a conserved active site with a catalytic triad consisting of a Cys, Gln, and Trp (S1 Fig) [57]. Based on homology, all of the *S. cerevisiae* and *C. albicans* GPxs are predicted to function as phospholipid peroxidases, like Gpx4 in humans, indicating that they are capable of directly acting on organic peroxides to reduce them to much more stable alcohols [56,61]. The *C. albicans* GPxs

A

| GPX Name | CGD Gene Name | Other Aliases | Assembly 19/21 Identifier | Systematic Name |
|---|---|---|---|---|
| GPX3 | GPX3 | GPX2, GPX4 | Orf19.4436 | C1_07350C |
| GPX31 | (none) | GPX1, GPS2, HYR1 | Orf19.86 | C6_00850W |
| GPX32 | GPX2 | GPX1, GPX3, GPS1 | Orf19.85 | C6_00840W |
| GPX33 | GPX1 | GPX3, GPS3 | Orf19.87 | C6_00860W |

B

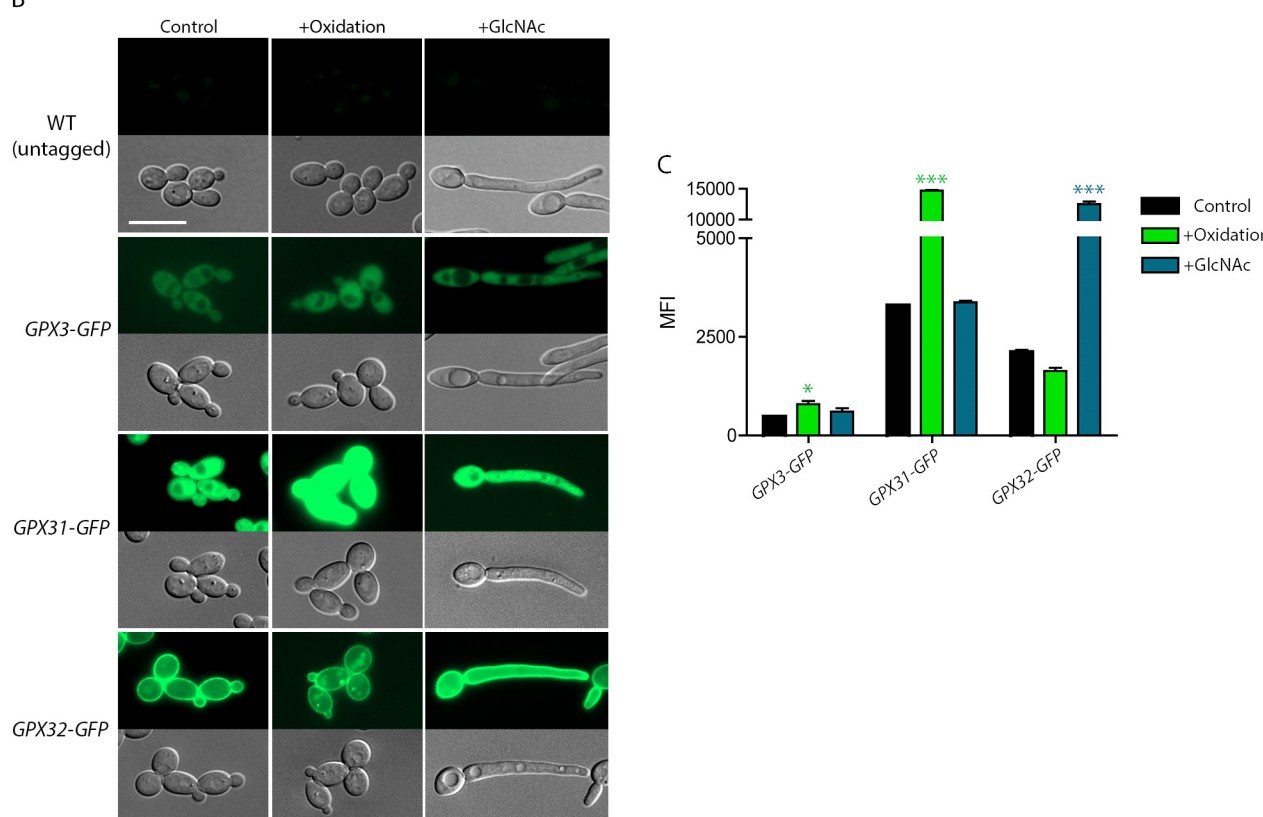

**Fig 2. *C. albicans* GPxs localize to the plasma membrane and cytoplasm.** (A) *C. albicans GPX* nomenclature for this study. (B) Fluorescence image of an untagged WT strain and the three indicated Gpx proteins tagged with GFP. Both alleles of each *GPX* gene were tagged with GFP to increase the fluorescent signal. Cells were grown at 30°C in media only (Control), at 30°C with 0.3 mM tert-butyl hydroperoxide (t-BHP) for 60 m (+Oxidation), or at 37°C with 50 mM GlcNAc for 120 m to induce hyphae (+GlcNAc). Representative microscope images shown correspond to the results of three independent experiments. The scale bar is 10 μm. (C) Quantification of the Gpx-GFP signals from the experiment described in (B). All images were taken with the same microscope settings. The mean fluorescence intensity (MFI) was calculated for each tagged protein under each condition. Fluorescence intensity was measured at the cytoplasm for Gpx3 and Gpx31, and at the plasma membrane for Gpx32. The microscopy images were processed under the same conditions for each GFP-tagged strain. However, due to the wide variation in the fluorescence intensity between the different GFP-tagged proteins, the images were adjusted to enable weaker GFP signals to be visualized. Statistical analysis of Gpx-GFP levels used one-way ANOVA with Tukey's multiple comparison test for each strain individually. * $p < 0.05$, ** $p < 0.01$, and *** $p < 0.001$.

have been referred to by many different aliases, but for this paper we will refer to them as *GPX3*, *GPX31*, *GPX32*, and *GPX33* (Fig 2A) as described [36,57].

To establish the production and localization of the *C. albicans* GPxs, we genetically tagged all four with Green Fluorescent Protein (GFP) and analyzed them by fluorescence microscopy (Fig 2B and 2C). Gpx33-GFP was not readily detectable, consistent with previous RNAseq studies showing it is expressed at a low level [3,62]. The other Gpx-GFP strains were imaged after being grown under control conditions or exposure to oxidative stress with tert-butyl hydroperoxide (t-BHP). t-BHP is not directly produced during the oxidative burst, but it is

commonly used as a representative organic peroxide in ROS studies [40,57]. In addition, cells were induced with N-acetylglucosamine (GlcNAc) to form hyphae [63,64]. Gpx3-GFP localized to the cytoplasm and the levels were similar across all conditions tested, with a small increase in the fluorescence signal when exposed to oxidation. Gpx31-GFP also localized to the cytoplasm but was highly induced by exposure to t-BHP. Gpx32-GFP had a more distinctive localization to the plasma membrane, but Gpx32 is not predicted to contain a transmembrane domain, so how Gpx32-GFP localizes to the plasma membrane is unknown. Gpx32-GFP remained in the plasma membrane when treated with t-BHP but also displayed some localization to intracellular puncta. Interestingly, Gpx32-GFP was highly induced in hyphae. In control studies, Western blots confirmed that the full-length Gpx-GFP proteins were produced, indicating that the cytosolic GFP signals were not due to free GFP proteolytically cleaved off the Gpx proteins. Other control studies confirmed that GFP tagging did not alter the sensitivity of the cells to oxidative stress (S2 Fig).

## *GPX3* is the main contributor to oxidative stress resistance

Previous studies in *S. cerevisiae* showed that the GPxs contribute to oxidative stress resistance and that mutant strains in which all three of the *GPX* genes had been deleted were more sensitive to $H_2O_2$ and organic peroxides [56]. Likewise, in *C. albicans*, previous studies indicated that a *gpx3Δ/Δ* mutant strain is more sensitive to $H_2O_2$ and that a *gpx31,32,33Δ/Δ* ("*GPX3* only") deletion mutant strain is sensitive to both $H_2O_2$ and organic peroxides [36,57]. However, a quadruple deletion mutant strain lacking all four *C. albicans GPX* genes was not previously created to establish the total contribution of the GPxs. Furthermore, the *gpx3Δ/Δ* studies were carried out with only $H_2O_2$ and not organic peroxides. Therefore, to determine the contribution of the GPxs to oxidative stress resistance, *GPX* deletion mutant strains were created using transient CRISPR Cas9 methods [65]. A *gpx* quadruple deletion mutant (*gpxΔ/Δ/Δ/Δ*) was created in two steps, since *GPX31*, *GPX32*, and *GPX33* are all adjacent on chromosome 6 in *C. albicans* and can be deleted in one step. Using disk diffusion halo assays (Fig 3A), the susceptibility of the *gpxΔ/Δ/Δ/Δ* mutant strain was compared to a wild-type (WT) strain and to a *cat1Δ/Δ* mutant strain, since Cat1 (catalase) is the most well-known peroxidase (Fig 3B and 3C). We found that the *gpxΔ/Δ/Δ/Δ* strain was significantly more sensitive to both $H_2O_2$ and the organic peroxide cumyl hydroperoxide (CHP). Similar to t-BHP, CHP is used in some assays as a representative organic peroxide [53]. CHP is more hydrophobic than t-BHP, so CHP was used in agar plate assays and t-BHP was used in liquid culture assays. The halo assays indicated that the *cat1Δ/Δ* mutant had a small increase in sensitivity to $H_2O_2$, compared to WT, but surprisingly the trend was not significant. As expected, the *cat1Δ/Δ* strain was not more sensitive to CHP. These results confirm that specific enzymes are required for detoxifying organic peroxides.

We next determined the contribution of each individual GPx protein to protection from ROS. Since the *GPX*s are highly homologous to each other, they could possibly have compensatory roles. We therefore created strains in which three of the *GPX* genes were deleted and only one *GPX* remained. Disk diffusion halo assays with $H_2O_2$ and CHP showed that only *GPX3* measurably contributes to oxidative stress resistance (Fig 3D and 3E). Similar results were observed at 30°C (Fig 3) and at the physiological temperature of 37°C (S3 Fig). To directly examine the accumulation of ROS, the cells were treated with t-BHP for 60 m and then the dye 2',7'-dichlorodihydrofluorescein diacetate (H2DCFDA) was added. This dye becomes fluorescent when oxidized and is used as a measure of intracellular ROS [55,66], as shown for the *gpxΔ/Δ/Δ/Δ* strain in Fig 3F. All *GPX* deletion strains lacking *GPX3* displayed increased ROS accumulation and a strain carrying *GPX3* as the only *GPX* gene in the cell

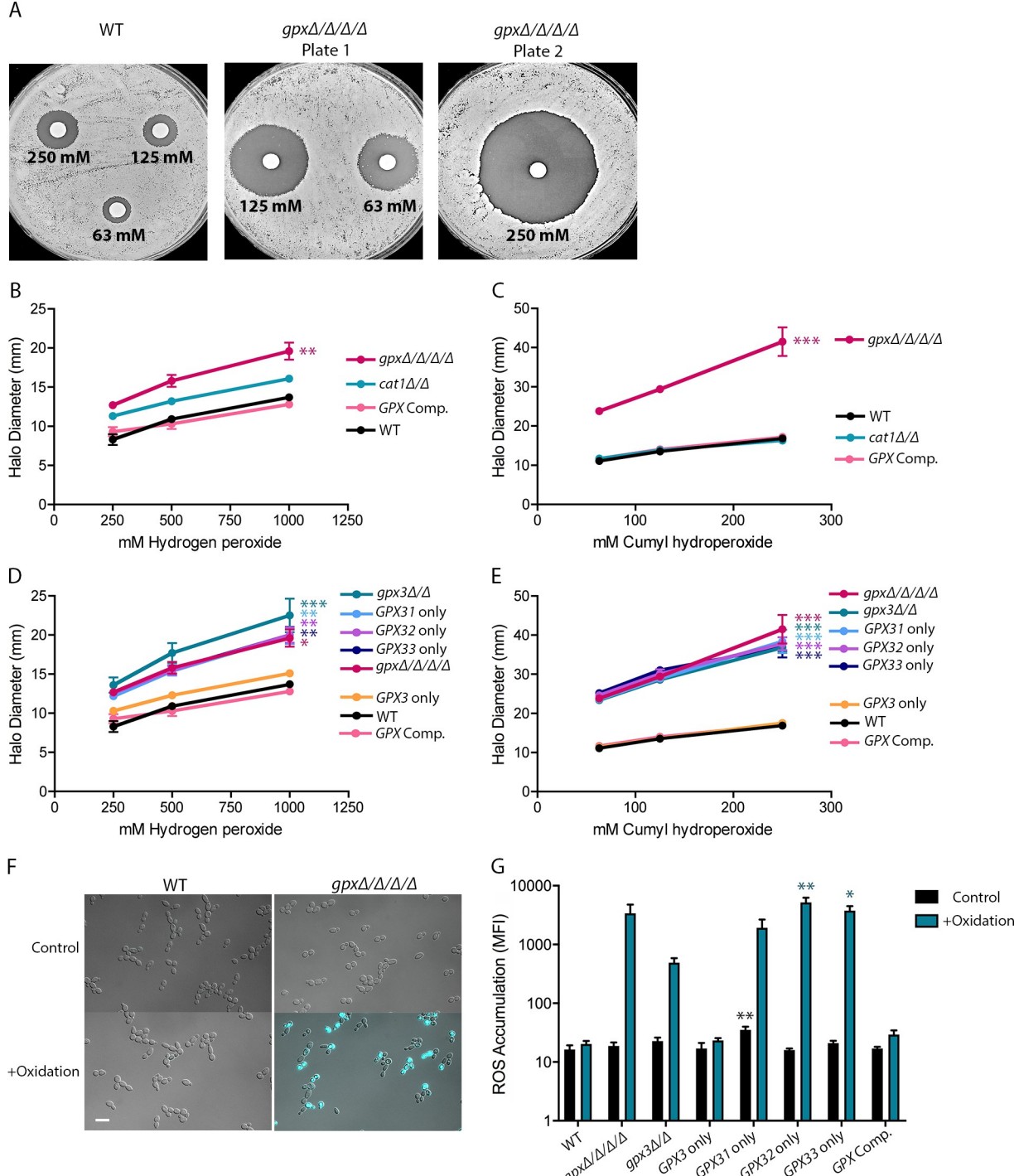

**Fig 3. *GPX3* plays the major antioxidant role of the GPxs in *C. albicans*.** (A) Representative images of a disk diffusion halo assay measuring the sensitivity to CHP for the WT and *gpxΔ/Δ/Δ/Δ* strains. The relative sensitivities of the *gpxΔ/Δ/Δ/Δ*, *cat1Δ/Δ*, *GPX* Comp., and WT strains using disk diffusion halo assays with (B) H₂O₂ and (C) CHP. Sensitivity of *gpx* deletion mutant strains to (D) H₂O₂ and (E) CHP. "*GPX* only" strains indicate a strain where three of the four *GPX*s were deleted and only the indicated *GPX* remains. For all halo assays, plates were incubated at 30°C and the diameter of the zone of inhibition was measured after 48 h. Results represent the average of three independent assays. Asterisks indicate strains with statistically significant different values at all concentrations when compared to WT. Strains with no asterisks had no significant differences from WT. (F) 2',7'-dichlorodihydrofluorescein diacetate (H₂DCFDA) assay to measure intracellular ROS accumulation. Cells were grown in YPD media alone (Control) or with 0.3 mM t-BHP (+Oxidation) for 60 m, stained with H₂DCFDA, and then imaged. The results represent 3–4 independent assays per strain and condition. Representative images of the WT and *gpxΔ/Δ/Δ/Δ* strain

are shown. The scale bar is 10 μm. (G) Quantification of the images of each *gpx* mutant strain after the treatment and staining in (F). Statistical analysis for halo assays and ROS accumulation used one-way ANOVA with Tukey's multiple comparison test. For the ROS accumulation assay, statistical significance was determined between values for each treatment condition. * $p<0.05$, ** $p<0.01$, and *** $p<0.001$.

(*GPX3* only) showed a low level of ROS similar to the WT control strain (Fig 3G). Due to the wide variation of the fluorescence signal of the dye in the different strains, ROS accumulation was only found to be statistically significant for the *GPX32* and *GPX33* only strains after treatment with t-BHP. However, it was interesting that the *gpx3Δ/Δ* mutant had a trend towards less ROS accumulation than the *gpxΔ/Δ/Δ/Δ* strain or the strains carrying only *GPX31*, *GPX32*, or *GPX33*. This result suggests that *GPX31*, *GPX32*, and *GPX33* may have some contribution to short-term oxidative stress resistance, although the phenotype is very weak.

### GPX deletion phenotype is due to loss of Cap1 activation

Previous studies in *C. albicans* found that in response to oxidative stress, Gpx3 promotes the formation of a disulfide bond in Cap1 that leads this transcription factor to accumulate in the nucleus and upregulate many important antioxidant genes [36]. This mechanism of activation is conserved in *S. cerevisiae* [31–35]. Since *GPX3* was the only *C. albicans* GPX with a significant antioxidant contribution, we wanted to determine if the phenotype for the *gpxΔ/Δ/Δ/Δ* and *gpx3Δ/Δ* strains was a result of a failure of the GPxs to act on peroxides or if it was possibly due to a defect in Cap1 activation. To test this, we conducted a genetic test in which we compared the phenotypes of a *gpxΔ/Δ/Δ/Δ* strain, a *cap1Δ/Δ* deletion mutant strain, and a mutant strain in which all four *GPX*s and *CAP1* were deleted (*gpxΔ/Δ/Δ/Δ+cap1Δ/Δ*). If the GPxs played a direct role in detoxifying peroxides, we expected the *gpxΔ/Δ/Δ/Δ+cap1Δ/Δ* strain to be more sensitive to oxidative stress than either the *cap1Δ/Δ* or *gpxΔ/Δ/Δ/Δ* strains. However, if the GPxs predominantly act to activate Cap1 then we expected the *gpxΔ/Δ/Δ/Δ+cap1Δ/Δ* mutant to mirror the *cap1Δ/Δ* phenotype. The results confirmed the latter hypothesis as the phenotypes were similar (Fig 4A and 4B). Surprisingly, this indicates that the *C. albicans* GPxs do not play a significant role in oxidative stress resistance via direct detoxification of peroxides. Rather, the primary antioxidant role of the GPx proteins is to activate Cap1. We also compared the *gpxΔ/Δ/Δ/Δ* and *cap1Δ/Δ* strains sensitivity at 37˚C and found the same trends as at 30˚C (S3 Fig). Interestingly, the *gpxΔ/Δ/Δ/Δ* and *gpx3Δ/Δ* mutants are significantly less sensitive than *cap1Δ/Δ* when exposed to $H_2O_2$ but have a similar level of sensitivity towards CHP (Figs 3D, 3E, 4A and 4B). This finding indicates that Gpx3 is the primary activator of Cap1 upon exposure to organic peroxides, but not upon exposure to $H_2O_2$.

We next wanted to eliminate the possibility that Cap1 in turn regulates the expression of the *GPX* genes. To our knowledge, this has not been found in previous transcriptomic and chromatin immunoprecipitation (ChIP) studies with Cap1 [37–39], but to verify these findings we tagged the *GPX* genes with GFP in a *cap1Δ/Δ* background strain (Fig 4C). There were no significant changes under control conditions for production of GFP-tagged GPx proteins between the WT and *cap1Δ/Δ* strains (Fig 4D). Although Gpx31-GFP and Gpx32-GFP were produced at lower levels in the *cap1Δ/Δ* strain after exposure to t-BHP for 60 m, this change could be due to the highly increased ROS sensitivity of *cap1Δ/Δ* affecting protein production. Nonetheless, the *GPX3* only strain did not display a significant phenotype, so even if *GPX31*, *GPX32*, or *GPX33* were regulated by Cap1, it would still not account for the lack of a synergistic phenotype between *cap1Δ/Δ* and *gpxΔ/Δ/Δ/Δ*. There was no significant change for Gpx3 production after treatment with t-BHP for 60 m. However, due to the high levels of autofluorescence in the *cap1Δ/Δ* background strain and the relatively low levels of Gpx3 production across all strains, we carried out shorter term assays in which cells were treated with t-BHP for

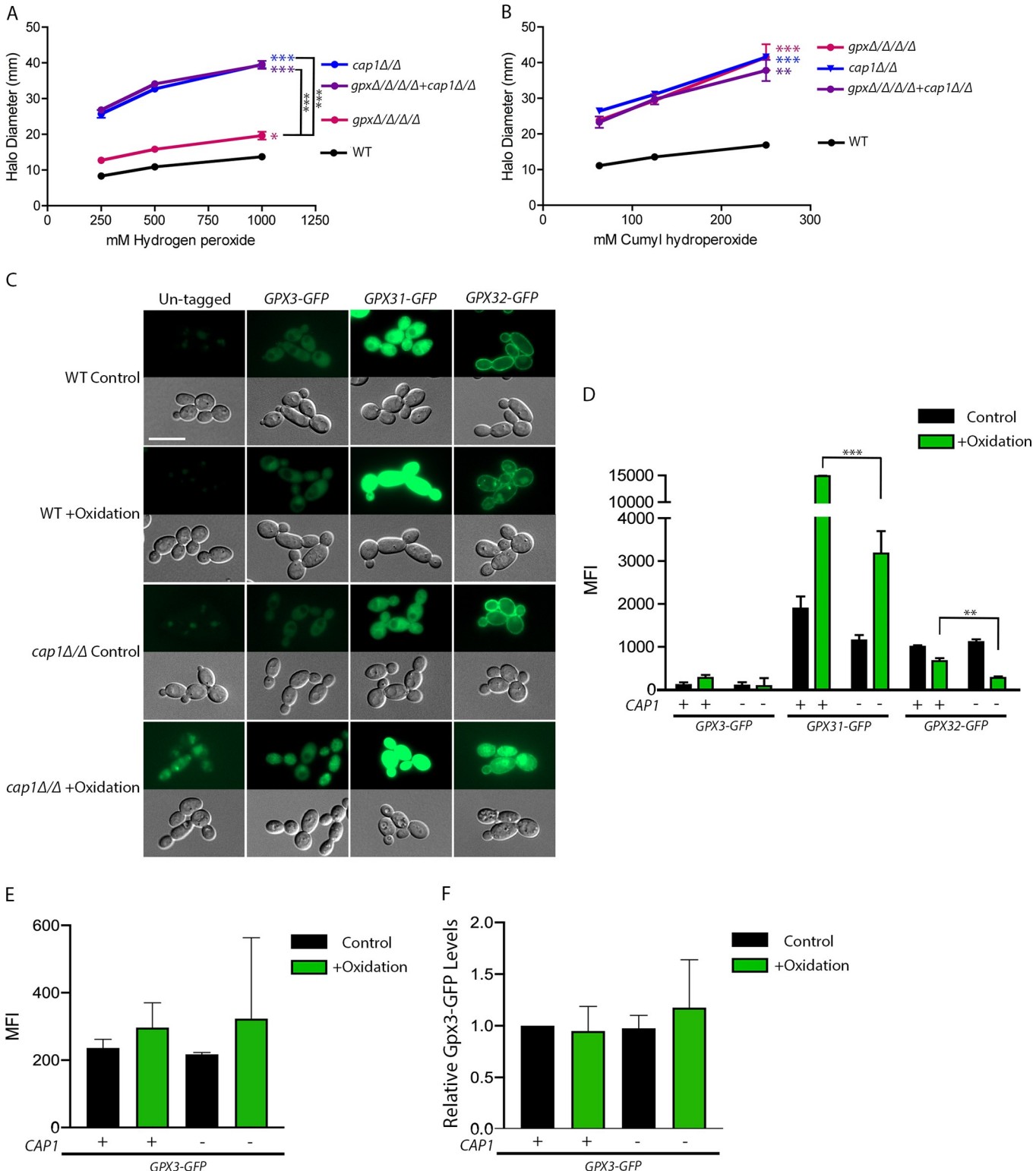

**Fig 4. Genetic relationships between *GPX* and *CAP1* mutations.** Relative sensitivities of the *gpxΔ/Δ/Δ/Δ*, *cap1Δ/Δ*, *gpxΔ/Δ/Δ/Δ+ cap1Δ/Δ*, and WT strains were assessed using disk diffusion halo assays with (A) $H_2O_2$ or (B) CHP at 30˚C. The diameter of the zone of inhibition was calculated after 48 h. The results represent 3–4 independent assays. For (A) and (B), colored asterisks indicate strains with significantly different values at all concentrations when compared to WT. Strains with no asterisks had no significant differences from WT. Brackets with black asterisks indicate select strains of interest with significant differences between all values. (C) Localization and intensity of GFP tagged GPxs in WT and *cap1Δ/Δ* strains. Cells were grown in YPD media alone (Control) or with 0.3

mM t-BHP for 60 m (+Oxidation). The scale bar is 10 μm. (D) Quantification of the images described in (C). (E) 30 m induction with *GPX3-GFP* strains. All microscopy data represents 3–4 independent assays per strain and condition. For (D) and (E), each column was normalized to their respective parental strain and condition. (F) Quantification of Western blots measuring Gpx3-GFP protein levels. Protein levels were normalized to Coomassie gels with whole cell lysates and are shown relative to WT untreated Gpx3-GFP levels. The reported values represent 4 independent assays. Brackets with black asterisks denote significant differences in MFI or Relative GFP Levels between the Gpx-GFP signals in the WT and *cap1Δ/Δ* deletion strains. No bracket indicates that no significant differences were found. Altogether, the results indicate that the antioxidant effects of the GPxs are due to Cap1 regulation and not direct detoxification of peroxides. All strains in this figure only had a GFP tag on one *GPX* allele. Fluorescence intensity was measured at the cytoplasm for Gpx3 and Gpx31. Fluorescence intensity was measured at the plasma membrane for Gpx32. Note that the brightness of all images was edited identically except for the *GPX31-GFP* "WT +Oxidation" image, in which *GPX31-GFP* is very highly induced. Statistical analysis for halo assays and Western blots used one-way ANOVA with Tukey's multiple comparison test. For D and E, statistical analysis for fluorescence quantification used Student's t-test to compare each strain and condition in the *cap1Δ/Δ* background to their respective WT counterpart. * $p < 0.05$, ** $p < 0.01$, and *** $p < 0.001$.

30 m which resulted in less autofluorescence in the *cap1Δ/Δ* strain than the 60 m treatment (Fig 4E). Once again, there was no significant change in Gpx3 levels across all conditions and strain backgrounds. To confirm these results, Western blot analysis was used to show that there were no significant differences in Gpx3-GFP protein levels between WT and *cap1Δ/Δ* strains, or between untreated samples and samples exposed to t-BHP for 30 m (Figs 4F and S4). These data demonstrate that the GPx proteins are still produced in the *cap1Δ/Δ* mutant and further confirm the hypothesis that the primary role of the GPxs in *C. albicans* is activation of Cap1. In agreement with this, studies in *S. cerevisiae* also found that expression of the ortholog of Gpx3 (also called Gpx3), which activates the *S. cerevisiae* Cap1 counterpart (Yap1) was not altered upon deletion of *YAP1* [60].

## Antioxidant pathways are highly redundant

ROS sensitivity assays showed that the *cap1Δ/Δ* strain is highly sensitive to $H_2O_2$ and organic peroxides, indicating that proteins important for the detoxification of peroxides are downstream of Cap1 regulation. Previous studies on Cap1 in *C. albicans* used microarrays and ChIP to identify the genes that are upregulated by Cap1. From this set, we selected *CAT1*, *TSA1*, *TRR1*, *TRX1*, and *GLR1* for further study, making sure that both the glutathione and thioredoxin antioxidant systems were represented [37–39]. As stated previously, *CAT1* (catalase) is the most well-known peroxidase and is important for $H_2O_2$ resistance in vitro, although it is not important for virulence in mice [67]. *TRR1* is a thioredoxin reductase, which maintains the reduced form of thioredoxin in the cell. *TRR1* is essential, so we created a heterozygous *TRR1* mutant (*trr1Δ*). The *TRX1* gene encodes a thioredoxin protein, which contains conserved catalytic Cys residues that allow it to function as an electron donor. The Trx1 protein is believed to reduce many antioxidant proteins including Tsa1 and the fungal GPxs [68–70]. *TSA1* is a thioredoxin peroxidase, or peroxiredoxin, meaning it detoxifies peroxides and relies on thioredoxin as an electron donor. In *C. albicans*, *TSA1* has a homolog, *TSA1B*, with an identical gene sequence [68,71,72]. Our mutant was therefore a double mutant with both genes deleted (*tsa1Δ/Δ+tsa1bΔ/Δ*). *GLR1* encodes a glutathione reductase, which reduces oxidized glutathione (GSSG) to reduced glutathione (GSH). GSH can then go on to promote reduction of ROS via numerous glutathione utilizing enzymes [70,73,74].

Surprisingly, although all of these genes have been linked to oxidative stress resistance, the mutant strains displayed at most slight increases in sensitivity to peroxides, none of which were statistically significant (Fig 5A and 5B). ROS accumulation was also evaluated using $H_2DCFDA$ staining. While not statistically significant, the *glr1Δ/Δ* mutant showed a trend of increased levels of ROS accumulation at 60 m of exposure to t-BHP, even though the *glr1Δ/Δ* mutant was not significantly more sensitive to oxidation in the 48 h halo assay (Fig 5C). These results indicate that *C. albicans* has evolved multiple redundant pathways to counteract oxidation.

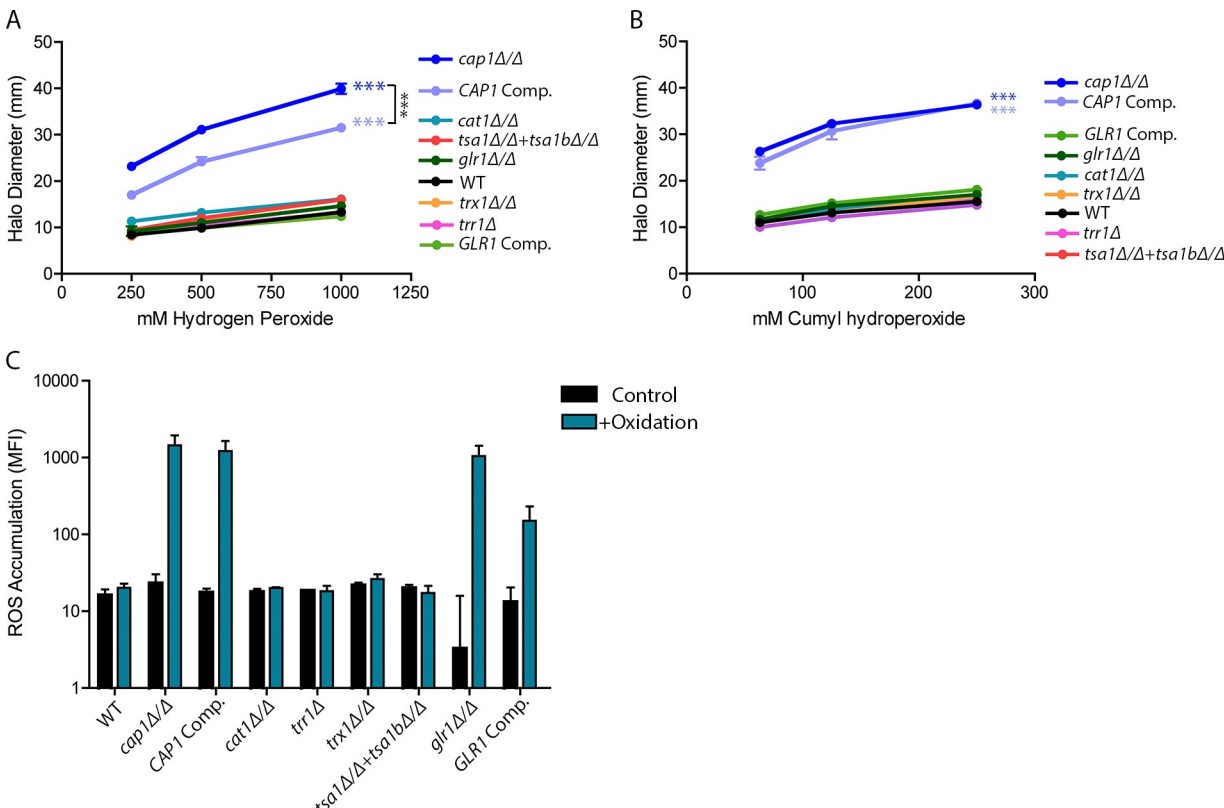

**Fig 5. Contribution of genes upregulated by Cap1 to oxidative stress resistance is likely combinatorial.** To assess the sensitivity of mutant strains lacking key antioxidant genes regulated by Cap1, disk diffusion halo assays were conducted with (A) $H_2O_2$ or (B) CHP at 30˚C. The diameter of the zone of inhibition was measured after 48 h. Results represent 3–4 independent assays. For (A) and (B), colored asterisks indicate strains with significantly different values at all concentrations when compared to WT. Strains with no asterisks had no significant differences from WT. Brackets with black asterisks indicate select strains of interest with significant differences between all values. (C) $H_2$DCFDA assay to measure intracellular ROS accumulation in mutant strains. Cells were grown in YPD media alone (Control) or with 0.3 mM t-BHP (+Oxidation) for 60 m, stained with $H_2$DCFDA, and then imaged. All microscopy assay data represents 3–5 independent assays per strain and condition. The MFI for each indicated strain and condition was determined. No significant differences between strains were found for ROS accumulation. Statistical analysis for halo assays and ROS accumulation used one-way ANOVA with Tukey's multiple comparison test. For the ROS accumulation assay, statistical significance was determined between values for each treatment condition. * $p < 0.05$, ** $p < 0.01$, and *** $p < 0.001$.

The *cap1Δ/Δ* mutant was complemented by reintroduction of *CAP1* at an ectopic location (NEUT5L), but this strain showed varying levels of rescue, with partial complementation being observed for $H_2O_2$ and at most very weak complementation observed for t-BHP or CHP (Fig 5). These results align with previous Cap1 research which also showed partial complementation of a *cap1Δ/Δ* mutant when *CAP1* was reintroduced [75]. This could be due in part to only one copy of *CAP1* being reintroduced or that *CAP1* may not be properly expressed at an ectopic location. To further ensure that our *cap1Δ/Δ* mutant strain was representative, we compared it to all four *cap1Δ/Δ* deletion strains present in the Homann and Noble transcription factor mutant libraries and found no significant differences in sensitivity (S5 Fig) [76,77].

## Cap1 and Glr1 are important for resistance to lipid peroxidation

Lipid peroxidation is an especially damaging effect of ROS exposure as it can start a chain reaction that spreads the oxidative damage to other lipids [44–47]. Therefore, we assessed the susceptibility of selected antioxidant mutant strains to lipid peroxidation by spotting serial

dilutions of cells onto agar plates containing linolenic acid (LNA (18:3)), oleic acid (OA (18:1)), or no added fatty acids (Fig 6A). LNA (18:3) is often used as a reagent when measuring sensitivity to lipid peroxidation because this PUFA has three conjugated double bonds, making it highly susceptible to autooxidation. In contrast, OA (18:1) serves as a control because this monounsaturated fatty acid is not highly susceptible to lipid peroxidation. Thus, mutant strains with impaired antioxidant responses to oxidized lipids should have increased sensitivity to LNA (18:3) but not OA (18:1) [53,78]. The results showed that the *gpxΔ/Δ/Δ/Δ*, *gpx3Δ/Δ*, *gpxΔ/Δ/Δ/Δ+cap1Δ/Δ*, *cap1Δ/Δ*, and *glr1Δ/Δ* strains were all highly sensitive to LNA (18:3). The *cat1Δ/Δ* strain showed no sensitivity, and the *CAP1* Comp. and *trx1Δ/Δ* strains showed limited sensitivity at the higher dose of LNA (18:3). However, spot assays are not quantitative, so to determine the magnitude of these mutant phenotypes we created a quantitative assay to measure sensitivity to lipid peroxidation. Due to the highly hydrophobic nature of LNA (18:3), it does not diffuse on the surface of an agar plate and cannot be used in disk diffusion halo assays. Instead, we developed an assay where 3 μL aliquots of dilutions of an LNA (18:3) solution were spotted onto a lawn of cells on an agar plate. After incubation for 24 h at 30°C to allow the lawn of cells to grow, the lowest concentration of LNA (18:3) that inhibited growth was recorded (Fig 6B). Control studies showed that OA (18:1) was not inhibitory at these concentrations (Fig 6A).

Using this LNA (18:3) dose-response assay we found that, similar to the CHP and t-BHP assays, all *gpx* mutant strains lacking *GPX3* had similar sensitivity to LNA (18:3) as the *gpxΔ/Δ/Δ/Δ* strain. Additionally, the strain expressing only *GPX3* (and not *GPX31*, *GPX32*, or *GPX33*) grew similar to WT (Fig 6C). Once again, the *gpxΔ/Δ/Δ/Δ*, *gpxΔ/Δ/Δ/Δ+cap1Δ/Δ*, and *cap1Δ/Δ* strains all had similar phenotypes indicating that, contrary to expectations, the *C. albicans* GPxs do not play a major role in detoxifying lipid peroxides (Fig 6D). These results differ from research in *S. cerevisiae* which found that the *yap1Δ* (*CAP1* ortholog) deletion mutant had no increased sensitivity to LNA (18:3) and that the *S. cerevisiae* GPxs do have a role for detoxifying oxidized lipids outside of Yap1 activation [79]. Furthermore, our results strongly indicate that proteins downstream of Cap1 are very important for protecting cells from lipid peroxidation. Specifically, our assays show that the glutathione reductase (*GLR1*) gene is important for protecting cells from lipid peroxidation. In mammals, GPxs are predicted to rely on glutathione reductases so that they can be reduced via GSH after being oxidized [80]. However, in yeast, the thioredoxin Trx1 has been shown to predominantly carry out the role of GPx reduction [81,82]. Furthermore, oxidized Gpx3 is required for Cap1 oxidation, not the reduced form of the enzyme. This makes it unlikely that the phenotype of the *glr1Δ/Δ* strain is due to a lack of reduced GPxs [33,70].

## Hog1 protects cells from oxidized lipids, but not other forms of oxidative stress

The other major stress response gene regulator in *C. albicans* is the Hog1 SAPK. Hog1 is well established to protect cells from osmotic and oxidative stress [40,41]. To define the role of Hog1 in protecting against organic peroxides, we assayed a *hog1Δ/Δ* deletion mutant strain, as well as the SAPKK *pbs2Δ/Δ* and SAPKKK *ssk2Δ/Δ* deletion mutant strains that lack upstream activators of Hog1 (Fig 7A and 7B). The mutants showed a slight trend towards increased sensitivity to $H_2O_2$ compared to WT strains, which was not statistically significant, and no change in sensitivity to CHP compared to WT. Interestingly, the Hog1 pathway deletion mutant strains were very strongly sensitive to LNA (18:3) (Fig 7C). This increased sensitivity to oxidized lipids but not to other peroxides indicates a special role for Hog1 in protecting against lipid peroxidation. One possibility is that LNA (18:3) causes more lipid peroxidation than

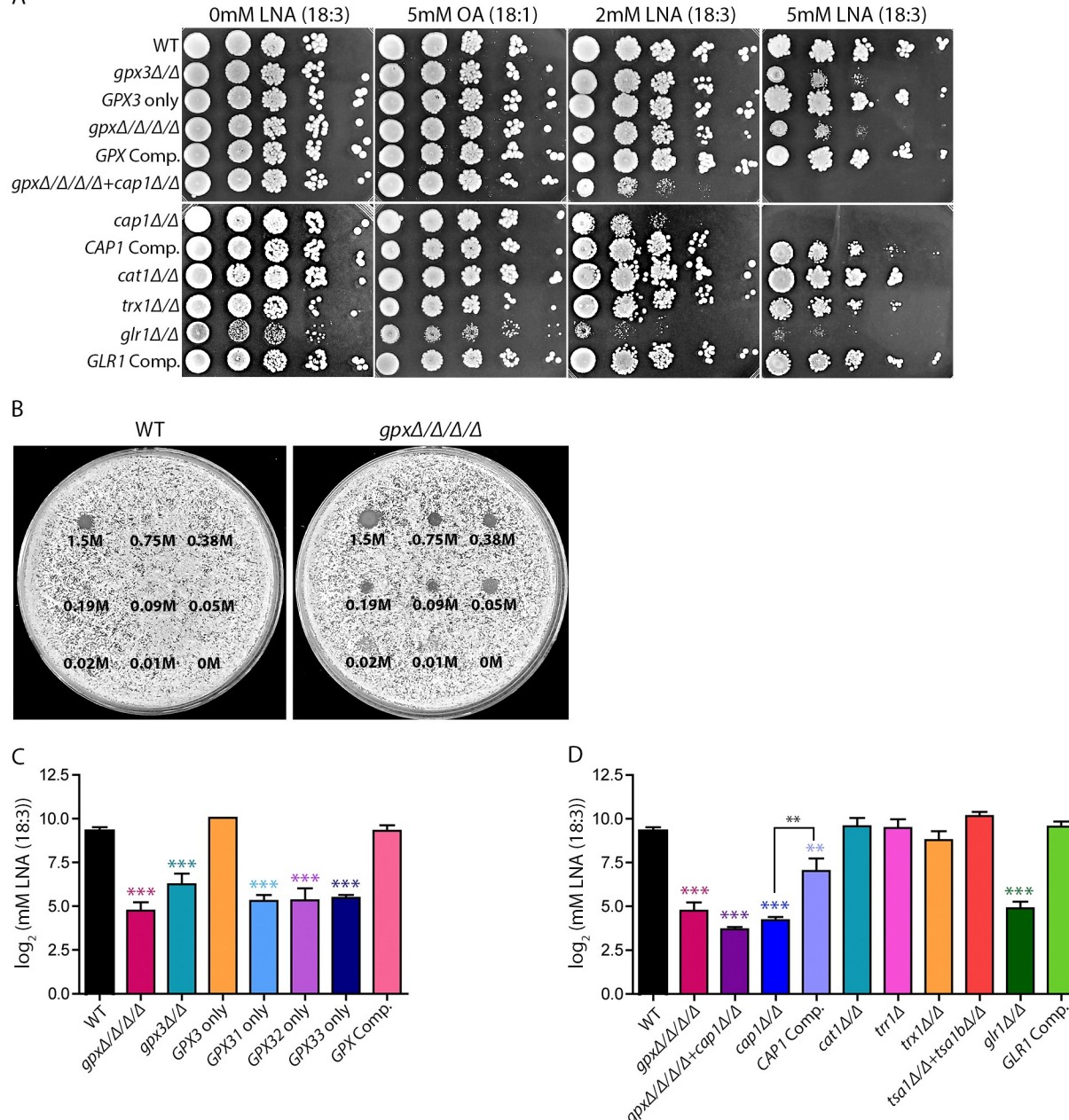

**Fig 6. Glutathione utilizing pathways induced by oxidative stress are important for resistance to lipid peroxidation.** Assays to determine sensitivity to lipid peroxidation. (A) Spot assay for which cells were grown on agar plates containing Linolenic acid (LNA (18:3)), Oleic acid (OA (18:1)), or DMSO only ("0 mM LNA (18:3)") for 48 h at 30°C and then imaged. (B) LNA (18:3) dose response assay to determine relative sensitivity to LNA (18:3). Dilutions of LNA (18:3) were spotted onto a lawn of cells and incubated for 24 h at 30°C before imaging. Representative images are shown. Based on the assay shown in (B), quantification of LNA (18:3) sensitivity for the (C) *GPX* mutants and (D) *cap1Δ/Δ* and related mutants. All LNA (18:3) assays represent a minimum of 3 independent experiments. Assays were quantified using ImageJ software to determine the LNA (18:3) concentration at which the lawn of cells fills in the site of the LNA (18:3) droplet. For (C) and (D), colored asterisks indicate strains with significantly increased sensitivity compared to WT. Strains with no asterisks had no significant differences from WT. Brackets with black asterisks indicate select strains of interest with significant differences in sensitivity. Statistical analysis for assays used one-way ANOVA with Tukey's multiple comparison test. * $p < 0.05$, ** $p < 0.01$, and *** $p < 0.001$.

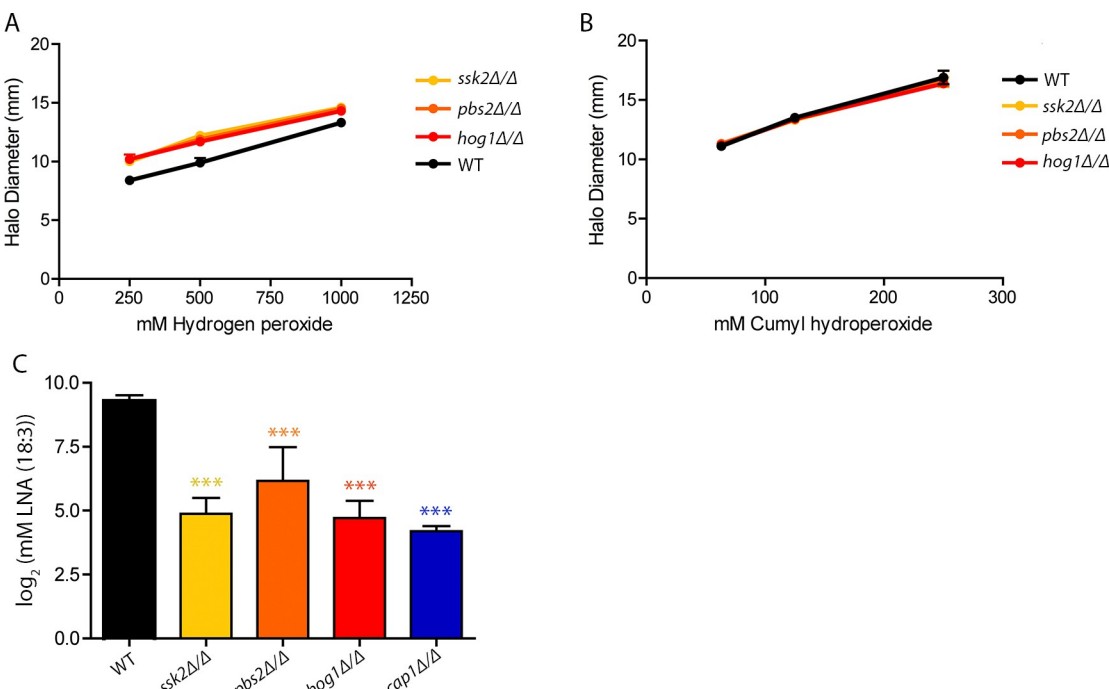

**Fig 7. Major stress response regulator, Hog1, plays a lesser role in oxidative stress resistance but promotes survival against LNA (18:3).** Sensitivity of *hog1Δ/Δ*, *pbs2Δ/Δ*, and *ssk2Δ/Δ* strains to oxidative stress. Disk diffusion halo assays were conducted with (A) $H_2O_2$ or (B) CHP at 30°C. The diameter of the zone of inhibition was calculated after 48 h. Results represent 3–4 independent assays. (C) LNA (18:3) dose response assay described in Fig 6 was used to determine the relative sensitivity of the Hog1 pathway mutants. The *cap1Δ/Δ* strain was included for comparison. The LNA (18:3) assays represent a minimum of 3 independent experiments for each strain. For all assays, asterisks indicate significant differences in sensitivity compared to WT. Strains with no asterisks had no significant differences from WT. In (C), there were no significant differences in sensitivity between the four mutant strains. Statistical analysis for assays used one-way ANOVA with Tukey's multiple comparison test. * p<0.05, ** p<0.01, and *** p<0.001.

either CHP or $H_2O_2$, leading to higher levels of damaged membrane lipids that alter the integrity of the plasma membrane causing leakage and increased osmotic stress. A *hog1Δ/Δ* strain may be defective in adapting to this kind of damage [41].

## Transcriptional responses to oxidative stress are highly conserved

The transcriptional response of *C. albicans* to $H_2O_2$ has been assessed with RNA-sequencing (RNAseq) previously [3,62]. However, there has not been a comprehensive RNAseq analysis of the *C. albicans* transcriptional response to organic peroxides. We therefore conducted RNAseq to determine the transcriptional response to 15 m of exposure to the organic peroxide t-BHP and found that 745 genes were upregulated ≥ 2-fold and 564 genes were downregulated ≥ 2-fold (Fig 8A). GO term analysis indicated that antioxidant pathways were the most upregulated by t-BHP, as expected, and both *GLR1* and the glutathione dependent antioxidant *GST1* were among the most highly induced genes (Fig 8B and 8C). For all genes upregulated ≥ 2-fold, protein degradation pathways were highly represented, indicating that organic peroxides may cause extensive cellular damage by oxidizing proteins (Fig 8D).

For comparison, RNAseq was also performed with cells treated with $H_2O_2$ for 15 m. The results showed that 359 genes were upregulated ≥ 2-fold and 236 genes were downregulated ≥ 2-fold in the t-BHP treated cells versus the $H_2O_2$ treated cells (Fig 9A). Of the top 10 most differentially regulated genes between the two conditions, no clear trends

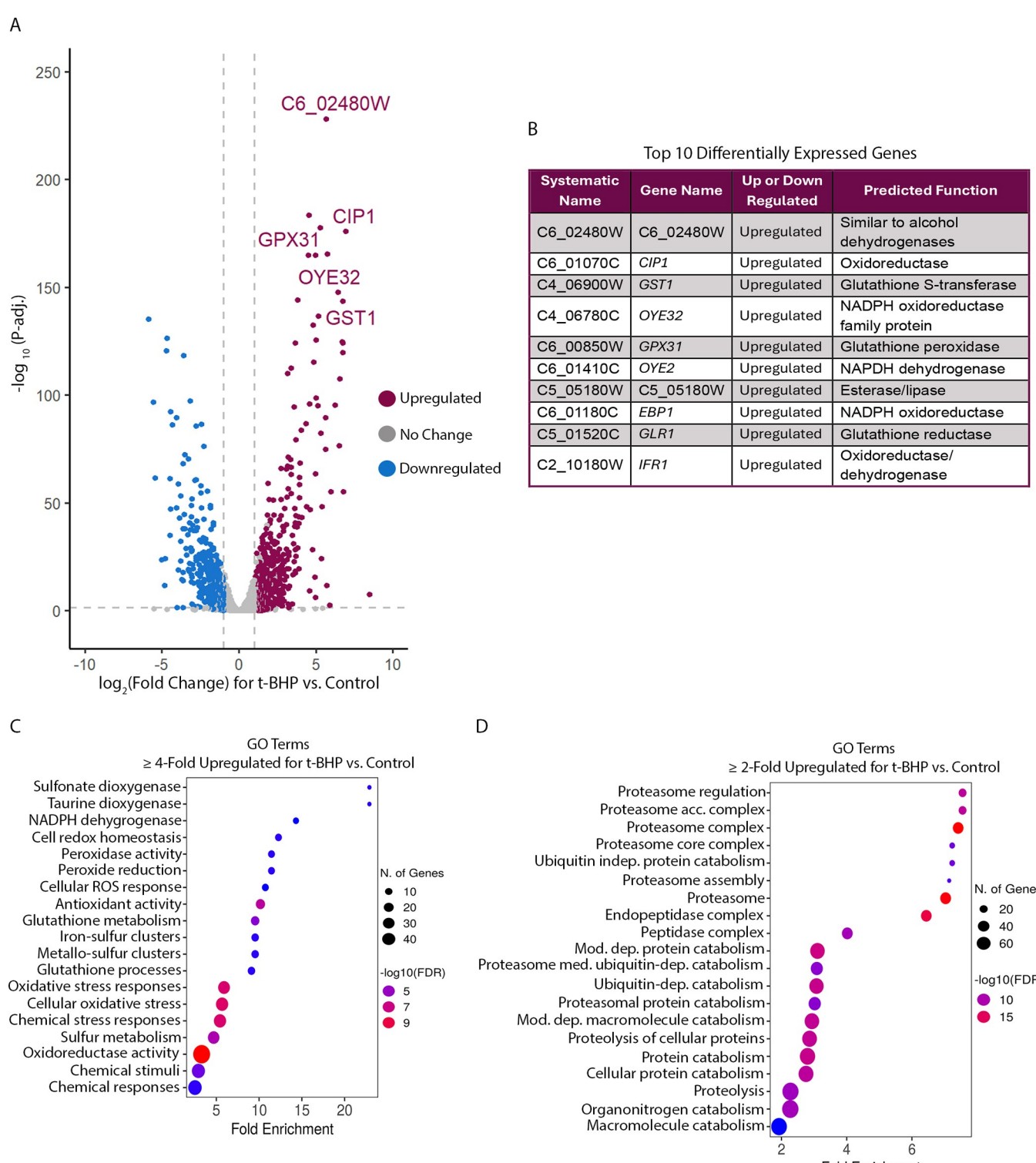

**Fig 8. Transcriptional response to exposure to the organic peroxide, t-BHP.** (A) Volcano plot of transcriptional changes for cells exposed to t-BHP compared to control conditions. The top 5 differentially expressed genes are indicated. (B) The top 10 differentially expressed genes. Differential expression was determined by multiplying the $\log_2$ (Fold Change) of gene expression by the $-\log_{10}$ (P-adj.) value for each gene. (C) GO terms for genes upregulated $\geq$ 4-fold upon exposure to t-BHP. (D) GO terms for genes upregulated $\geq$ 2-fold upon exposure to t-BHP. Results for each condition represent 3 independent RNAseq experiments. Only transcriptional responses with P-adj. < 0.05 were used in the GO term analyses.

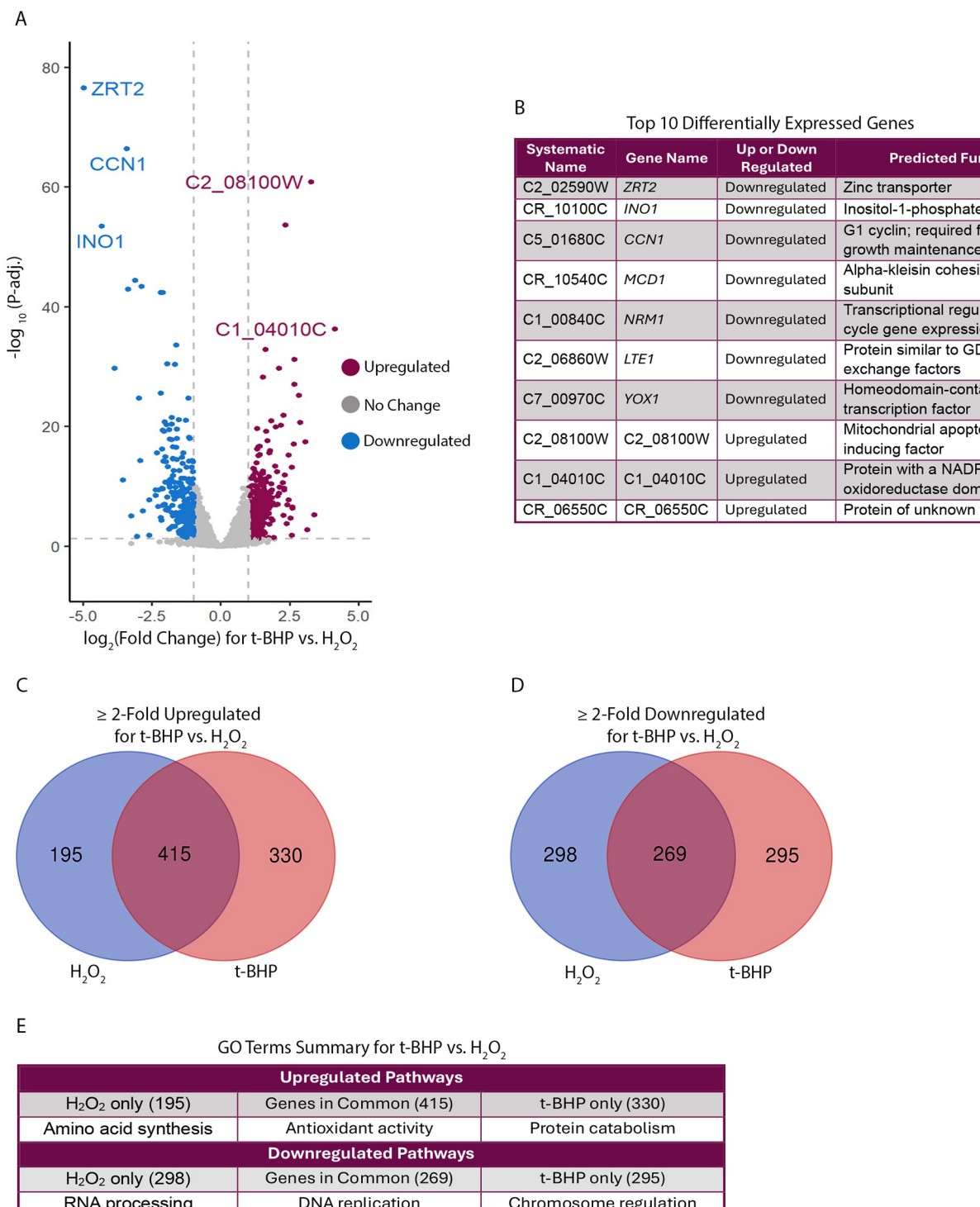

**Fig 9. The transcriptional responses to t-BHP and $H_2O_2$ show a conserved oxidative stress response.** (A) Volcano plot for transcriptional changes in cells exposed to t-BHP compared to $H_2O_2$. The top 5 differentially expressed genes are indicated. (B) The top 10 differentially expressed genes. Differential expression was determined by multiplying the $\log_2$ (Fold Change) of gene expression by the $-\log_{10}$ (P-adj.) value for each gene. (C) Venn diagram of genes with a $\geq$ 2-fold upregulated transcriptional response for t-BHP or $H_2O_2$. (D) Venn diagram of genes with a $\geq$ 2-fold downregulated transcriptional response for t-BHP or $H_2O_2$. (E) Summary of the most dominant GO term category for each Venn diagram grouping. Results for each condition represent 3 independent RNAseq experiments. Only transcriptional responses with P-adj. < 0.05 were used in the Venn diagram and GO term analyses.

emerged (Fig 9B). GO term analysis for genes upregulated $\geq$ 2-fold compared to control conditions showed that both $H_2O_2$ and t-BHP induced pathways involved in antioxidant activity. However, $H_2O_2$ caused upregulation of genes involved in amino acid synthesis and t-BHP led to upregulation of genes involved in protein catabolism (Fig 9C and 9E). GO term analysis for genes downregulated $\geq$ 2-fold compared to control conditions showed that both $H_2O_2$ and t-BHP downregulated genes involved in DNA replication, t-BHP downregulated genes involved in chromosome regulation, and $H_2O_2$ downregulated genes involved in RNA processing (Fig 9D and 9E). These findings support reports of cell cycle arrest when *C. albicans* is exposed to oxidative stress [83]. More detailed GO terms analyses are included in S6 and S7 Figs. Thus, although the effects of $H_2O_2$ and the organic peroxide, t-BHP, were overlapping, there were significant differences consistent with these oxidants having distinct chemical properties.

## Discussion

Pathogens must have efficient stress response mechanisms to successfully infect a host. For most fungal pathogens, the host innate immune system is the primary line of defense against infection. To kill invading pathogens, phagocytic immune cells generate superoxide which is very reactive and can directly damage pathogens or be converted into other ROS such as peroxides and hypochlorous acid [18,20]. However, most previous studies on oxidative stress responses in fungal pathogens, including *C. albicans*, have examined $H_2O_2$ as a model ROS. Therefore, we wanted to determine how *C. albicans* combats organic peroxides and lipid peroxidation. This is an understudied area of research but is important as 30% of the fatty acids in *C. albicans* are PUFAs, making it susceptible to lipid peroxidation [2,52,53]. Our initial studies examined the *C. albicans* GPxs since they were predicted from studies in mammalian cells and *S. cerevisiae* to have a key role in protecting against lipid peroxidation and detoxifying other organic peroxides in *C. albicans* [28,57,58]. Surprisingly, genetic studies indicate that activation of Cap1 is the major role of the GPxs, not the direct detoxification of peroxides.

*GPX3* was the only one of the four *GPX* genes in *C. albicans* with a measurable role in oxidative stress resistance (Figs 3D, 3E, 3G and 6C). This role for Gpx3 was not because it was the most abundant GPx, as GFP-tagging experiments showed that Gpx3 is produced at lower levels than Gpx31 or Gpx32 (Fig 2B and 2C). Furthermore, Gpx3-GFP was not strongly induced by oxidation while Gpx31-GFP was highly induced (Figs 2B, 2C and 4C–4F). If the impact of GPxs on oxidative stress resistance was due to direct detoxification of peroxides, we would expect Gpx31 and Gpx32 to be the most important due to their much higher protein levels, especially since the active sites are conserved between all four *C. albicans* GPxs. Furthermore, the *gpxΔ/Δ/Δ/Δ* strain did not show synergy between the GPxs, as it had the same phenotype as the *gpx3Δ/Δ* strain (Figs 3D, 3E, 3G, 6A and 6C). This conclusion was also supported by data showing that the strain expressing *GPX3* as the only *GPX* had similar levels of ROS resistance as the WT control strain. These data indicate that *GPX3* has an important antioxidant role other than direct detoxification of peroxides.

Gpx3 is known to regulate the transcription factor Cap1, which raised the possibility that the main role of Gpx3 is antioxidant gene regulation [36,58]. During oxidative stress, Gpx3 promotes a conformational change in Cap1 that leads to nuclear accumulation of Cap1 and increased expression of antioxidant genes (Fig 1A) [24,36,38,39]. Consistent with the main role of Gpx3 being Cap1 activation, the *cap1Δ/Δ*, *gpx3Δ/Δ*, and *gpxΔ/Δ/Δ/Δ* mutant strains showed very similar levels of sensitivity to organic peroxides (Figs 3D, 3E, 4B and 6). Furthermore, a mutant strain lacking all four *GPX* genes and *CAP1* (*gpxΔ/Δ/Δ/Δ+ cap1Δ/Δ)* did not show synergistic effects. If the antioxidant action of the GPxs was independent of Cap1 regulation, we would expect an increase in oxidative stress sensitivity in the *gpxΔ/Δ/Δ/Δ+ cap1Δ/Δ*

strain compared to the *cap1Δ/Δ* strain. Instead, the *gpxΔ/Δ/Δ/Δ+ cap1Δ/Δ* and *cap1Δ/Δ* strains displayed similar susceptibility towards the oxidants we tested, including LNA (18:3) (Figs 4A, 4B and 6D). This contrasts with results in *S. cerevisiae* where a *yap1Δ* mutant was not more susceptible to LNA (18:3) [60]. Our results do not rule out a direct role for the GPxs in detoxifying peroxides under other conditions or as a backup pathway [79]. But, they do show that the major function of the *C. albicans* GPxs is to activate Cap1.

Interestingly, we also found that Gpx3 is the primary activator of Cap1 upon exposure to organic peroxides, including lipid peroxides, but not to $H_2O_2$ (Figs 4A, 4B, 6A, 6C and 6D). A previous study in *C. albicans* found that the Ybp1 protein stabilizes the reduced form of Cap1 in the cytoplasm, which allows it to be more easily oxidized by Gpx3 [36]. It is possible that Ybp1 alone can promote some activation of Cap1 under certain conditions. Furthermore, studies in *S. cerevisiae* implicate a thioredoxin dependent peroxidase (homologous to Tsa1 in *C. albicans*) that acts as a backup system for Cap1 activation [36,84]. Future work should further elucidate Cap1 activation under various oxidative stress conditions.

To better define the role of Cap1, we examined key genes upregulated by Cap1 to determine their role in preventing lipid peroxidation. In addition to their impact on membrane architecture, lipid peroxides can also degrade into toxic downstream products including aldehydes [45,48], indicating a need for diverse antioxidant proteins that can detoxify a range of ROS and oxidized molecules. Analysis of selected mutants from both the thioredoxin and glutathione systems showed that the thioredoxin system mutants did not exhibit significantly increased sensitivity to lipid peroxidation; however, the glutathione reductase (Glr1) mutant was highly sensitive to oxidized lipids (Fig 6A and 6D). Glutathione reductase utilizes NADPH to reduce GSSG to GSH, which is important for the function of many GSH dependent enzymes. Future studies should focus on defining the role of glutathione utilizing enzymes in detoxifying oxidized lipids, including the glutaredoxins, which are small proteins that reduce oxidized molecules via thiol reactions. There are four glutaredoxins in *C. albicans*, one of which was found to be essential for virulence [85–87]. Another interesting group of glutathione utilizing enzymes is the large family of glutathione S-transferase (GST) proteins, many of which have not yet been fully characterized [88,89]. GSTs promote GSH binding to hydrophobic oxidized molecules, such as lipids, which increases the solubility of the oxidized compound, making it possible to export the compound or degrade it intracellularly [85,90]. Interestingly, our RNAseq data showed very significant upregulation of *GST1* in response to oxidative stress (Fig 8B).

To our knowledge, we conducted the first RNAseq assay of the transcriptional response of fungal cells to an organic peroxide. The results showed that *C. albicans* has a generalized oxidative stress response (Figs 8B, 8C and 9E). Many key antioxidant genes were upregulated in both the $H_2O_2$ and t-BHP conditions, even though some specific gene products were ineffective against certain types of ROS. For example, *CAT1* was highly induced by t-BHP treatment, even though Cat1 cannot act on organic peroxides. Similarly, previous studies showed that *CAT1* was induced by benzoquinone and hypochlorous acid, which are oxidizing agents that also cannot be acted on by catalase [3]. These findings are consistent with previous reports of a generalized ROS response in *C. albicans* [3,62]. However, there were some noticeable differences in the transcriptional responses to $H_2O_2$ and t-BHP. One of the main differences was an increase in protein catabolism pathways in t-BHP treated cells (Fig 9E), suggesting that t-BHP may be more toxic to proteins, leading to an increased need for protein catabolism.

We also found that Hog1 pathway mutants showed significantly increased sensitivity to lipid peroxidation, but not other forms of oxidative stress, indicating that this phenotype of the Hog1 pathway mutants was not due to a general susceptibility to oxidation (Fig 7). Previous studies reported that Hog1 does not significantly affect the general antioxidant transcriptional

response [40,41]. These data suggest an alternative role for Hog1 in oxidative stress resistance outside of transcriptional regulation of antioxidant genes. One possibility is Hog1 activated stress response pathways, such as the osmotic stress response, help to mitigate cellular damage after oxidation of membrane lipids weakens the integrity of the plasma membrane. Future studies will help define this important new role for Hog1.

Overall, this study established key differences between *C. albicans* antioxidant pathways and those in mammals and *S. cerevisiae*, including the differing roles for GPxs and Cap1 in protecting against lipid peroxidation (Figs 3, 4 and 6). These findings demonstrate the importance of studying antioxidant pathways in pathogenic fungi and not relying solely on homology to model organisms [24]. Our results also highlight the importance of studying a range of different oxidants, and not just $H_2O_2$. In this regard, our findings synergize with the previous discovery in *C. albicans* of a family of FLPs that act as quinone reductases and are important for protecting against lipid peroxidation [53]. Combining the information from these studies, we developed a new model for *C. albicans* pathways that protect against the oxidation of membrane lipids (Fig 10). This model proposes that FLPs reduce ubiquinone to ubiquinol so that it can reduce lipid radicals, and that GPxs act primarily to activate Cap1 to induce a wide range of antioxidant genes that detoxify a broad range of ROS. In addition, the HOG pathway is proposed to help counter membrane damage resulting from lipid peroxidation. These antioxidant mechanisms identified in *C. albicans* have significance for developing novel therapeutic strategies that can synergize with current antifungal therapies that act on plasma membrane lipids and proteins [91]. Future studies examining this should investigate the impact of membrane lipid composition on susceptibility to oxidation and define the effects of oxidative stress on different membrane lipids. The results of our study also have significance for better understanding lipid peroxidation mechanisms in ferroptosis, which has been gaining interest as a strategy for cancer therapy [48,49,92]. Ferroptosis has recently been identified in fungi as a mechanism that regulates the virulence of plant fungal pathogens and is now a burgeoning topic in fungal research [93–96].

## Materials and methods

### Strains and media

The genotypes and short-hand names for the strains used in this study are described in S1 Table. Strains were kept on YPD agar plates (yeast extract, 2% peptone, 2% dextrose). For all agar plate assays, cells were grown on synthetic minimal medium agar plates containing 2% dextrose, 80mg/L uridine, and amino acids (SC+uri). For liquid cultures, cells were grown in rich YPD medium (2% yeast extract, 1% peptone, 2% dextrose, 80 mg/L uridine) or a synthetic minimal medium containing yeast nitrogen base, 2% dextrose, and uridine (SD+uri) [97]. Unless otherwise specified, all assays were carried out at 30°C.

A GFPγ tag was fused onto the 3' end of the open reading frame (Orf) of the *GPX* genes. The GFP sequence was PCR amplified from the pFA-GFPγ plasmid using primers described in S2 Table and introduced to *C. albicans* cells (strain SN152) via electroporation using 0.2 cm gene pulser electroporation cuvettes (Bio-Rad, Hercules, CA) [98]. Correct incorporation of the GFP tag into the genome was confirmed by PCR and fluorescence microscopy.

Deletion strains were created using transient CRISPR Cas9 methods which have been previously described [65]. In brief, a 20 bp guide RNA sequence targeted the CRISPR Cas9 enzyme to cut sites in the Orf of the target genes. Healing fragments with flanking regions of ~80 bp of homology to the directly adjacent 5' and 3' ends of the Orf were incorporated into the *C. albicans* genome via homologous recombination. This process led to a full excision of the entire Orf of the gene of interest. Healing fragments were constructed using the pSN69, pSN40, or

## A. Lipid peroxidation chain reaction.

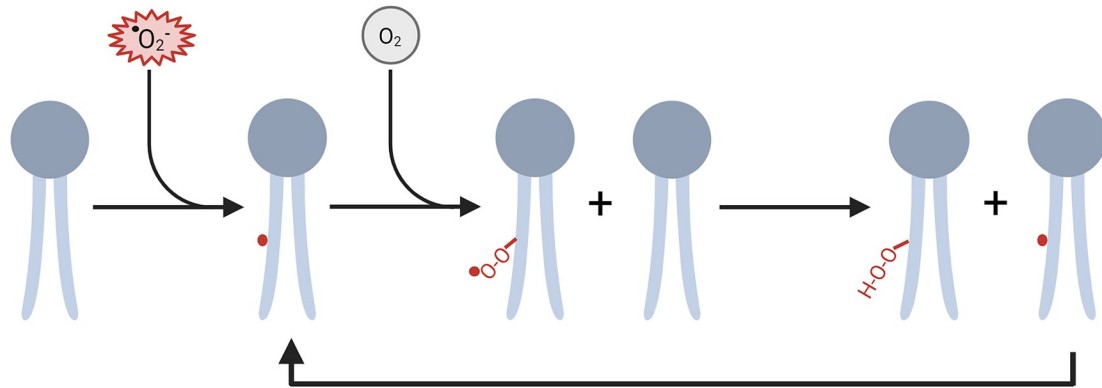

## B. Cellular pathways that protect the plasma membrane.

**Fig 10. Model for how _C. albicans_ protects against lipid peroxidation.** (A) Chain reaction of lipid peroxidation. Lipid radicals react with molecular oxygen to form a lipid peroxyl radical. This lipid peroxyl radical then reacts with another membrane lipid to form a lipid peroxide and a new lipid radical. This reaction will repeat until quenched. (B) Pathways that protect the plasma membrane. Oxidative stress comes in many forms during infection, including superoxide ($O_2^-$), hypochlorous acid (HOCl), and metal ions ($Cu^{2+}$ and $Fe^{2+}$). These oxidants damage membrane lipids and form lipid radicals, lipid peroxides, and degraded lipids, which damage the integrity of the membrane. SODs act on superoxide to convert it to the less reactive, $H_2O_2$. Gpx3 activates the transcription factor Cap1, which upregulates expression of antioxidant genes, including _GLR1_. Glr1 maintains reduced glutathione levels in the cell, so glutathione utilizing proteins, such as the glutathione S-transferases (Gst) and the glutaredoxins (Grx), can act on oxidized lipids. Hog1 acts in both the cytoplasm and in the

nucleus where it activates pathways that counteract osmotic stress. The FLPs promote reduction of quinones, which in turn reduce and detoxify lipid radicals. Figure created using BioRender.com.

pSN52 plasmids described previously [99] and introduced into cells using electroporation. Deletions were confirmed via PCR using primers in the regions upstream and downstream of the targeted gene, primers within the Orf, and primers with homology to the respective healing fragments. 3–5 independent deletion mutants were identified and compared to confirm they had similar phenotypes.

Complementation of deletion strains was performed using methods described previously [100]. The WT gene to be complemented was PCR amplified along with ~1,000 bp upstream to include the promoter region and ~300 bp downstream to include the terminator of the gene. Primers used for amplification of the target gene (including the promoter, Orf, and terminator) contained 20 bp of homology to the target gene and 80 bp of homology to the pDIS3 plasmid. The pDIS3 plasmid was cut using the restriction enzyme *Sma* I and was then co-transformed into *S. cerevisiae* (strain W3031A) with the gene of interest to create a complementing plasmid via gap repair. The resulting plasmid, including the nourseothricin resistance gene (*NAT*) was PCR amplified and introduced to the respective mutant strain using electroporation. For the *GPX* Comp. strain, the *GPX3* gene was inserted into the pDIS3 plasmid using the methods described above. The resulting plasmid was transformed into electroporation competent DH5α *E. coli* cells. These cells were grown up and the pDIS3+*GPX3* plasmid was purified. The region directly downstream of GPX3 insertion in the pDIS3+*GPX3* plasmid was cleaved using *Cla* I and a segment of DNA spanning from ~1000 bp upstream of *GPX32* to ~300 bp downstream of *GPX33* was inserted via gap repair. The entire sequence containing all four *GPX* genes and all flanking regions (~10 kb) was then PCR amplified and introduced to the NEUT5L region of the *gpxΔ/Δ/Δ/Δ* strain, as described above. Correct insertion of all complemented sequences was confirmed with PCR.

### Microscopy

For all microscopy assays, liquid cultures were grown in YPD at 30˚C, unless otherwise specified. Microscope slides were coated with 1% UltraPure Agarose (Invitrogen, Waltham, MA) to prevent cell mobility during imaging. For GFP assays, all cultures were grown overnight and then diluted to $0.4 \times 10^7$ cells/mL and incubated for 2 h before any treatments were added. For yeast phase GFP assays, oxidative stress treated cells were grown for the indicated time with 0.3 mM t-BHP (Acros Organics BV, Geel, Belgium). Control cells were grown for the additional time with nothing added. For hyphal assays, 50 mM of GlcNAc was added and cultures were grown for an additional 120 m at 37˚C. After treatments, cells were spun down, washed once with water, and then resuspended in water before imaging.

The ROS accumulation assay used 2',7'-dichlorodihydrofluorescein diacetate ($H_2$DCFDA) (Invitrogen, Waltham, MA) suspended in dimethyl sulfoxide (DMSO, MP Biomedicals, Santa Ana, CA). Liquid cultures were grown overnight, diluted to $0.4 \times 10^7$ cells/mL, and incubated for 2 h. Control condition cells were grown for an additional hour with no treatment. Oxidative stress treated cells were grown for the additional hour with 0.3 mM t-BHP. $H_2$DCFDA was added to cultures to a final concentration of 10 μM. Cultures were then incubated for 20 m in the dark and then spun down, washed once, and resuspended in water before imaging.

The microscope used was a Zeiss Axio Observer 7 microscope equipped with a Zeiss AxioCam 702 digital camera. Zeiss ZEN software was used to quantify fluorescence intensity.

For all assays, the fluorescence intensity of cells was determined for 100 cells per strain and condition for a minimum of 3 biological replicates, resulting in a minimum of 300 cells total

per strain and condition. The mean fluorescence intensity (MFI) was then calculated by averaging those measured values.

## Halo assays

Overnight liquid cultures were grown in YPD. Cells were then diluted to $1 \times 10^6$ cells/mL. 250 μL of cells were spread onto SC+uri agar plates. When the surface of the agar was dry (about 30–60 m), 10 μL of solutions containing the indicated amount of $H_2O_2$ (Fisher Chemical, Hampton, NH) or CHP (Acros Organics BV, Geel, Belgium) mixed with water were pipetted onto 6 mm sterile paper disks (BBL, Becton Dickinson Limited, Franklin Lakes, NJ). Disks were then placed onto the lawn of cells and the plates were incubated at the indicated temperatures. All halo assays were done in duplicate each day to ensure the consistency of the results. Plates were imaged and the zone of growth inhibition was measured after 48 h incubation.

## Western blot

Western blot methods have been described previously [101]. A *SUR7-GFP* strain was used as a positive control and the *GPX-GFP* strains with both alleles tagged with GFP were used for S2 Fig while a *GPX3-GFP* single tag in both WT and *cap1Δ/Δ* backgrounds was used for Figs 4F and S4 [42]. Strains were grown overnight in 5 mL YPD. Next day, 150 μL of the overnight cultures were added to 10 mL YPD and grown for an additional 8 h. Samples for Figs 4F and S4 were then treated with 0.3 mM t-BHP for 30 m. Cells were spun down, supernatant was removed, and cell pellets were frozen at -80°C. Cell pellets were then thawed, resuspended in a lysis buffer (2% SDS, 10% glycerol, 125 mM Tris-HCl, pH 6.8), and zirconia beads were added. Cells underwent bead bashing for 1 m followed by 1 m on ice, which was repeated 4 times. Samples were then spun down, and the soluble fraction was transferred to a clean tube. 2-mercaptoethanol and bromophenol blue were added to the samples to final concentrations of 5% and 0.002%, respectively. Lysates were boiled at 99°C for 10 m. Samples were separated by SDS-PAGE on a 10% polyacrylamide gel before being transferred to a 0.2 μm nitrocellulose membrane (Amersham Protran, catalogue# 10600032) using a semidry transfer device. After blocking for 1 h, the nitrocellulose membrane was incubated with a monoclonal mouse anti-GFP antibody (catalogue# 632381 JL-8, Takara Bio, San Jose, CA) for 1 h, followed by washing with TBS-Tween buffer, and then incubation with a goat anti-mouse IgG antibody for 1 h (IRDye 680RD, LI-COR Biosciences, Lincoln, NE). The membrane was washed again with TBS-Tween buffer and then imaged by scanning with an Odyssey Clx infrared imaging system (LI-COR Biosciences). As a control, polyacrylamide gels with the same samples were stained with Coomassie Brilliant Blue (0.1% Coomassie R-250, 40% ethanol, 10% glacial acetic acid) for 6 h. Gels were destained overnight using a destaining solution (40% methanol, 10% acetic acid). Images for Figs 4F and S4 were quantified using ImageJ software. In short, the signal intensity for Gpx3-GFP was divided by the signal intensity of total protein levels from the respective lane of the Coomassie stained gel. Relative protein levels were then quantified. The ratio of Gpx3-GFP levels to total protein levels for the WT untreated strain from each replicate was set to 1.0 and all other strains and conditions were reported relative to that value.

## LNA (18:3) assays

For Linolenic Acid (LNA (18:3)) spot assays, molten SC+uri agar was mixed with Linolenic Acid, Oleic Acid, or DMSO to the indicated concentrations. Oleic acid (OA (18:1)) and DMSO were used as controls. DMSO was added at same volume of LNA (18:3) in the 5 mM LNA (18:3) plate. 5 mL of 10% tergitol in SD+uri media was also added to each solution. LNA

(18:3), OA (18:1), and tergitol were all supplied by Sigma Aldrich, St. Louis, MO. The final volume of all solution mixtures was 50 mL. 25 mL of the solution was poured into each petri plate (Kord-Valmark, Seattle, WA) and allowed to dry overnight. Overnight liquid cell cultures were grown in YPD. Cultures were then diluted to $1x10^7$ cells/mL and then 2 μL of a series of 10-fold dilutions of cells were spotted onto plates and allowed to dry. All assays were done in duplicate each day to ensure the consistency of the results. Plates were then incubated at 30°C for 48 h and imaged. More detailed methods for the plates are found in S3 Table.

For LNA (18:3) dose response assays, overnight cultures were diluted to $1x10^6$ cells/mL and 250 μL of the cell dilution was pipetted onto SC+uri agar plates. Once cells dried onto the agar surface, 3 μL of the indicated LNA (18:3) or OA (18:1) dilutions were pipetted onto plates. Plates were then incubated at 30°C for 24 h before being imaged. LNA (18:3) dilutions were made with LNA (18:3), 10% tergitol in SD+uri, SD+uri media, and DMSO. The highest concentration of LNA (18:3) used was 1.5 M. The 0 mM (Control) dilution contained DMSO equal to the volume of LNA (18:3) in the 1.5 M LNA (18:3) solution. A 1.5 M OA (18:1) droplet was tested as a control. More detailed methods for the dilutions are found in S4 Table.

Quantification of the LNA (18:3) dose response assays was conducted with ImageJ software. All images were converted to black and white and then the threshold value was set to the same level for all images, making areas with cell growth white (high light intensity) and areas with no growth black (low light intensity). Since the 1.5 M LNA (18:3) droplet was lethal for all strains, the lowest effective LNA (18:3) dose was calculated by first measuring the light intensity (LI) of the highest concentration (1.5 M LNA (18:3)) and lowest concentration (0 M LNA (18:3)) droplet areas. Second, the lowest effective dose was determined to be the lowest dose at which the light intensity was less than (LI of 0 M LNA (18:3)—LI of 1.5 M LNA (18:3))/2, or less than half the value of the difference between the lowest and highest doses.

## RNA-sequencing

To determine the correct concentrations for $H_2O_2$ and t-BHP treatments, CFU assays were conducted to define the peroxide concentrations that result in ~95% cell viability. The sample preparation protocol was described previously [3]. Briefly, cells were grown overnight in SD +uri media and then diluted via a 6-fold dilution series and incubated overnight. A cell culture in log phase was then diluted to $10^6$ cells/mL and incubated until cell concentration was $10^7$ cells/mL (about 7 h). 10 mL of the cell cultures were added to 3, 15 mL tubes and $H_2O_2$ or t-BHP were then added to their respective cultures at a final concentration of 0.5 mM $H_2O_2$ or 0.5 mM t-BHP. The third tube received no oxidant and was used as a control. Cultures were incubated for 15 m before being spun down at 4°C and washed with ice cold water. After the final spin, the supernatant was removed and cells were flash frozen with liquid nitrogen before being stored at -80°C. Frozen cell pellets were submitted to Azenta (South Plainfield, NJ, USA) for RNA-sequencing.

The extraction of RNA, synthesis of cDNA, and sequencing procedures were performed at Azenta. RNA was isolated from a frozen cell pellet containing $1x10^8$ cells using the RNeasy Plus Universal Mini Kit, following the manufacturer's guidelines (Qiagen, Germantown, MD). The RNA concentration was determined using a Qubit 2.0 Fluorometer (Life Technologies), and RNA integrity was checked with an Agilent TapeStation 4200 (Agilent Technologies, Santa Clara, CA). For sequencing, RNA samples were processed using the NEBNext Ultra II RNA Library Prep Kit for Illumina, adhering to the manufacturer's protocol (NEB, Ipswich, MA). Briefly, mRNA was enriched with Oligo(dT) beads, fragmented at 94°C for 15 m, and used to synthesize cDNA. The cDNA fragments were end-repaired, adenylated at the 3' ends, and ligated with universal adapters, followed by index addition and library amplification via

limited-cycle PCR. The sequencing libraries were validated using an Agilent TapeStation (Agilent Technologies), quantified using a Qubit 2.0 Fluorometer (Invitrogen), and assessed by quantitative PCR (KAPA Biosystems). The libraries were pooled, clustered on a single lane of a flowcell, and sequenced on an Illumina HiSeq instrument (4000 or equivalent) using a 2x150 bp Paired End (PE) configuration. Image analysis and base calling were performed using HiSeq Control Software (HCS). The raw sequence data (.bcl files) were converted to fastq files and de-multiplexed using Illumina's bcl2fastq 2.17 software, allowing one mismatch for index sequence identification. Subsequently, the sequence data underwent quality profiling, adapter trimming, read filtering, and base correction using fastp, an all-in-one FASTQ preprocessor (DOI: 10.1093/bioinformatics/bty560). The high-quality paired-end reads were aligned to the *C. albicans* SC5314 genome (Candida Genome Database; Assembly 22) using HISAT2 (DOI: 10.1038/nmeth.3317). The resulting read alignments were assembled with StringTie (DOI: 10.1038/nprot.2016.095) to estimate transcript abundances. Absolute mRNA abundance was expressed as fragments per kilobase of transcript per million mapped reads (FPKM). Differential gene expression analysis was conducted using the DESeq2 package (DOI: 10.1186/s13059-014-0550-8) from Bioconductor (DOI: 10.1186/gb-2004-5-10-r80) on R. To visualize RNA-sequencing data, volcano plots were generated using R Studio, and GO term plots were created with ShinyGO 8.0 software (DOI: 10.1093/bioinformatics/btz931) [102]. Venn diagrams were also created using the online tool (https://bioinformatics.psb.ugent.be/webtools/Venn/).

## Statistical methods

All assays were conducted in triplicate, at minimum. Statistical analysis was done with Graph-Pad Prism software. Statistical analysis with more than 2 samples was determined using one-way ANOVA with Tukey's multiple comparison test. For comparisons between only two samples, an unpaired Students t-test was utilized.

## Supporting information

**S1 Table. Strain names for all strains used in this study.**
(PDF)

**S2 Table. All primers used for strain creation in this study.**
(XLSX)

**S3 Table. LNA (18:3) Spot Assay.**
(PDF)

**S4 Table. LNA (18:3) Dose Response Assay.**
(PDF)

**S5 Table. RNAseq gene expression fold change for all genes and conditions.**
(XLSX)

**S6 Table. Genes for S6 Fig. GO Terms.**
(XLSX)

**S7 Table. Genes for S7 Fig. GO Terms.**
(XLSX)

**S8 Table. Numerical Data for all figures.**
(XLSX)

**S1 Fig. Sequence alignment for *C. albicans* GPxs.** Sequence alignment for the four *C. albicans* Gpx proteins, the three *S. cerevisiae* Gpx proteins, and the human Gpx4 protein. Residues highlighted in red mark the Cys, Gln, and Trp catalytic triad. Note that for *Homo sapiens* Gpx4, the catalytic triad contains a selenocysteine ("X") at residue 73 and not a Cys. Asterisks indicate completely conserved residues, a single dot indicates partially conserved residues, and a colon denotes residues with similar properties. Alignment was created using Clustal software (https://www.ebi.ac.uk/jdispatcher/msa/clustalo).
(TIF)

**S2 Fig. Evaluation of *GPX-GFP* tagged strains.** To determine if GFP tagged Gpx3 is functional, a *GPX3-GFP* strain tagged on both *GPX3* alleles was compared to WT and *gpx3Δ/Δ* strains to assess any increased sensitivity to oxidative stress. Disk diffusion halo assays were conducted with (A) $H_2O_2$ or (B) CHP at 30˚C with 3–4 independent assays carried out per strain. Asterisks indicate strains with statistically significant differences in values at all concentrations when compared to WT. Strains with no asterisks had no significant differences from WT. The *GPX3-GFP* strain was tested because the *gpx3Δ/Δ* mutant had a measurable phenotype. (C) Western blot analysis indicating that the Gpx-GFP fusion proteins were full-length and that there was little or no evidence of a free GFP tag that had been proteolytically cleaved. A *SUR7-GFP* strain was used as a positive control. Strains with both *GPX* alleles tagged with GFP were used. (D) Coomassie stained gel to compare relative protein levels for samples used in (C). The Western blots and respective Coomassie stained gels were completed in triplicate. Statistical analysis for halo assays used one-way ANOVA with Tukey's multiple comparison test. * $p < 0.05$, ** $p < 0.01$, and *** $p < 0.001$.
(TIF)

**S3 Fig. Oxidative stress sensitivity of key mutant strains at 37˚C.** To determine oxidative stress sensitivity of key mutant strains at physiological temperature, disk diffusion halo assays were conducted with (A) $H_2O_2$ or (B) CHP at 37˚C. The diameter of the zone of inhibition was calculated after 48 h. The results represent 3–4 independent assays. Colored asterisks indicate strains with significantly different values at all concentrations when compared to WT. Strains with no asterisks had no significant differences from WT. Brackets with black asterisks indicate select strains of interest with significant differences between all values. Statistical analysis for halo assays used one-way ANOVA with Tukey's multiple comparison test. * $p < 0.05$, ** $p < 0.01$, and *** $p < 0.001$.
(TIF)

**S4 Fig. Gpx3-GFP levels in WT and *cap1Δ/Δ* background strains.** Western blots were conducted for WT and *cap1Δ/Δ* strains that each carried a single copy of *GPX3-GFP*. (A) Western blot showing Gpx3-GFP levels in WT and *cap1Δ/Δ* strains with and without exposure to 0.3 mM t-BHP for 30 min. (B) Corresponding Coomassie gel showing the total protein levels of the same samples in (A). These are representative images from 4 independent assays.
(TIF)

**S5 Fig. Oxidative stress sensitivity of *cap1Δ/Δ* strains.** To determine the oxidative stress sensitivity of various *cap1Δ/Δ* strains, disk diffusion halo assays were conducted with (A) $H_2O_2$ or (B) CHP. Plates were incubated at 30˚C and the diameter of the zone of inhibition was measured after 48 h. The results represent 3–4 independent assays. (C) Quantitative assay for sensitivity to LNA (18:3). All assays compared the sensitivity of WT and five *cap1Δ/Δ* strains: the prototrophic *cap1Δ/Δ* strain, the two *cap1Δ/Δ* strains from the Homann collection, and the two *cap1Δ/Δ* strains from the Noble collection. Assays for each strain were carried out a minimum of 3 independent times. Asterisks indicate strains with significantly different values at all

concentrations when compared to WT. No significant differences in sensitivity were found between the five $cap1\Delta/\Delta$ strains in any of the assays. Statistical analysis for assays used one-way ANOVA with Tukey's multiple comparison test. * p<0.05, ** p<0.01, and *** p<0.001.
(TIF)

**S6 Fig. GO terms for t-BHP vs. H$_2$O$_2$ ≥ 2-fold upregulated.** GO terms analysis for comparing the t-BHP and H$_2$O$_2$ transcriptional responses. GO terms were determined for transcriptional responses that had a log$_2$(Fold Change) value of ≥ 1.00, which is equivalent to a ≥ 2-fold transcriptional upregulation when compared to control samples. (A) GO terms for ≥ 2-fold upregulated genes in common between the H$_2$O$_2$ and t-BHP treated samples. (B) GO terms for genes upregulated ≥ 2-fold upon exposure to H$_2$O$_2$, but not t-BHP. (C) GO terms for genes upregulated ≥ 2-fold with exposure to t-BHP, but not H$_2$O$_2$. Only transcriptional responses with P-adj. < 0.05 were used in the analysis.
(TIF)

**S7 Fig. GO terms for t-BHP vs. H$_2$O$_2$ ≥ 2-fold downregulated.** GO terms analysis comparing the t-BHP and H$_2$O$_2$ transcriptional responses. GO terms determined for transcriptional responses that had a log$_2$(Fold Change) value of ≤ -1.00, which is equivalent to a ≥ 2-fold transcriptional downregulation when compared to control samples. (A) GO terms for ≥ 2-fold downregulated genes in common between the H$_2$O$_2$ and t-BHP treated samples. (B) GO terms for genes downregulated ≥ 2-fold upon exposure to H$_2$O$_2$, but not t-BHP. (C) GO terms for genes downregulated ≥ 2-fold with exposure to t-BHP, but not H$_2$O$_2$. Only transcriptional responses with P-adj. < 0.05 were used in the analysis.
(TIF)

## Acknowledgments

We thank the members of our lab for their advice and their assistance with editing this manuscript.

## Author Contributions

**Conceptualization:** Kara A. Swenson, James B. Konopka.

**Data curation:** Kara A. Swenson.

**Formal analysis:** Kara A. Swenson, Kyunghun Min.

**Funding acquisition:** James B. Konopka.

**Investigation:** Kara A. Swenson.

**Methodology:** Kara A. Swenson, James B. Konopka.

**Project administration:** Kara A. Swenson, James B. Konopka.

**Resources:** James B. Konopka.

**Software:** Kyunghun Min.

**Supervision:** James B. Konopka.

**Validation:** Kara A. Swenson, Kyunghun Min, James B. Konopka.

**Visualization:** Kara A. Swenson, Kyunghun Min.

**Writing – original draft:** Kara A. Swenson.

**Writing – review & editing:** Kara A. Swenson, Kyunghun Min, James B. Konopka.

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
