## [Decision Letter · Decision Letter 0]

20 Sep 2024

Dear Dr Konopka,

Thank you very much for submitting your Research Article entitled 'Candida albicans pathways that protect against organic peroxides and lipid peroxidation.' to PLOS Genetics. The manuscript was fully evaluated at the editorial level and by independent peer reviewers. The reviewers appreciated the attention to an important topic but identified some concerns that we ask you address in a revised manuscript.

Please consider the editorial suggestions, especially the point about nomenclature by Rev 3.  Rev 1 suggested some gene regulation studies that would be quite interesting, and (if the AE had to guess) that are probably underway.  If they are in a state where they can be folded in, please do so; if they are intended for a future paper then we will all just have to wait - with bated breath!

To resubmit, log into your Editorial Manager account and select the option 'Revise Submission' in the 'Submissions Needing Revision' folder.

Yours sincerely,

Aaron P. Mitchell, PhD

Academic Editor

PLOS Genetics

Eva Stukenbrock

Section Editor

PLOS Genetics

Reviewer's Responses to Questions

**Comments to the Authors:**

Reviewer #1: This is a very interesting study on detoxification of organic peroxides in Candida albicans. The authors find that unlike other organisms, the primary role of glutathione peroxidases in C. albicans is regulation of a transcription factor (Cap1) for oxidative stress as opposed to direct reduction of organic peroxides. The authors for the first time identify the transcriptome for the organic peroxide response in C. albicans. The findings will be of great interest to readers in fields of microbial oxidative stress resistance. I only have one major and two minor suggestions for improvement.

Major

1) The RNAseq with t-BHP and H2O2 is interesting, but it was only done with WT cells, and without any tests for effects of GPX3, CAP or HOG mutations, it doesn't seem to integrate well into the rest of the study. To bring it all together, the authors should pick a few of the favorite hits from RNAseq and see how gpx3 (or other mutants) impact this regulation by t-BHP vs H202.

Minor

2) In Fig. 2, it would be nice to show the fluorescence of a no-GFP cell, to demonstrate that the low signal from the control GPX3-GFP results from GFP and not background.

3) Figs. 2C, 3F, 4E, 5C are in need of statistical analyses to determine significance. How many cells and experimental trials do these represent?

Reviewer #2: The production of reactive oxygen species is a key aspect of the antimicrobial activity of immune cells, particularly phagocytes. An extensive literature exists to demonstrate the importance of antioxidant mechanisms in a wide variety of pathogens, including Candida albicans. In C. albicans, this has been heavily focused on exposure to hydrogen peroxide and on the role of the classical defense mechanisms, catalase and superoxide dismutases, as well as a transcriptional regulator Cap1. This manuscript extends these studies in three interesting ways, via investigation of lipid oxidation, four glutathione peroxidase proteins, and the impact of organic oxidants on C. albicans. Key findings are that only one of the four glutathione peroxidase genes, GPX3, has a detectable impact on ROS sensitivity and this is indirect, via activation of Cap1. Further, the GPX genes are important for resistance to organic peroxides (like tBH), but less so for peroxide, while catalase, for instance, shows the opposite pattern. Phenotypes of a mutant lacking GLR1, a glutathione reductase, support the particular importance of glutathione in response to lipid oxidation. All of this suggests that cellular defenses against ROS differ based on the type of oxidant, or the type of damage. Given this conclusion, the considerable overlap in gene expression in response to tBH vs. hydrogen peroxide is a little surprising.

This is an important paper, and a welcome extension to our understanding of ROS resistance in this pathogen. The experiments are thorough and clear and justify the conclusions, some of which are a bit unexpected. The genetics, in particular, are comprehensive. Further, the paper is very well written. I believe this is appropriate for PLoS Genetics and did not identify any significant issues that need to be addressed.

Reviewer #3: In this manuscript by Swenson et al, the authors examined the role of four glutathione peroxidases (GPXs) on reducing organic peroxides in the cell. Most previous work on antioxidants in C. albicans had been done with hydrogen peroxide, but lipid peroxidation is an important physiological event. The authors determine the cellular localization of the GPxs upon organic peroxides stress. Using growth sensitivity and fluorescent superoxide accumulation assays upon various organic stresses they show that GPX3 is the major player, and that the major role of the GPxs is to activate the transcription factor Cap1. Assays employing the direct addition of fatty acids containing 1 (oleic) and 3 (linolenic) double bonds they show that Cap1 and Glr1 are important for resistance to lipid peroxidation. Finally, they perform RNAseq to identify the transcriptional response to t-BHT. The paper is well-written, the data are of high quality, and the the topic is important. A few minor concerns are noted below.

1. A number of figure legends, for example 3-5, don't have N numbers for number of halos assays performed, etc

2. Halo assay is quantified in several figures, but we don't see a picture. At least one picture of the assay would be appropriate.

3. I expect lipid biochemist to be interested in this paper. When discussing the lipid additions it might be useful to also use to the 18:1 and 18:3 nomenclature to describe oleic and linolenic acids- the reader then does not need to memorize names to get the key point.

4. Discussion and background points that would strengthen the paper if addressed:

Are there published lipidomic studies on the impact of oxidative stressors on lipids in C.a.? If so, are their any particular lipids that are impacted more than others? What are the major takeaways from those studies?

To solidify the importance of these results, it seems that lipidomic studies in the various mutant backgrounds are a logical next step? I did not see this addressed in the Discussion.

**Have all data underlying the figures and results presented in the manuscript been provided?**

Reviewer #1: Yes

Reviewer #2: Yes

Reviewer #3: Yes

PLOS authors have the option to publish the peer review history of their article (what does this mean?). If published, this will include your full peer review and any attached files.

Reviewer #1: No

Reviewer #2: No

Reviewer #3: No

---

## [Editor Report · Decision Letter 1]

8 Oct 2024

Dear Dr Konopka,

We are pleased to inform you that your manuscript entitled "Candida albicans pathways that protect against organic peroxides and lipid peroxidation." has been editorially accepted for publication in PLOS Genetics. Congratulations!

Yours sincerely,

Aaron P. Mitchell, PhD

Academic Editor

PLOS Genetics

Eva Stukenbrock

Section Editor

PLOS Genetics

Comments from the reviewers (if applicable):

**Data Deposition**

http://datadryad.org/submit?journalID=pgenetics&manu=PGENETICS-D-24-00929R1

**Press Queries**

---

## [Editor Report · Acceptance letter]

14 Oct 2024

PGENETICS-D-24-00929R1 

Candida albicans pathways that protect against organic peroxides and lipid peroxidation. 

Dear Dr Konopka, 

We are pleased to inform you that your manuscript entitled "Candida albicans pathways that protect against organic peroxides and lipid peroxidation." has been formally accepted for publication in PLOS Genetics! Your manuscript is now with our production department and you will be notified of the publication date in due course.

With kind regards,

Anita Estes

PLOS Genetics

On behalf of:
